# Melt at grounding line controls observed and future retreat of Smith, Pope, and Kohler Glaciers

David A Lilien[1,2], Ian Joughin[1], Benjamin Smith[1], and Noel Gourmelen[3]

[1]Applied Physics Laboratory, University of Washington, Seattle, WA, USA
[2]Department of Earth and Space Sciences, University of Washington, Seattle, WA, USA
[3]School of Geosciences, University of Edinburgh, Edinburgh, UK
*Correspondence to*: David A. Lilien (dal22@uw.edu)

**Abstract**

Smith, Pope, and Kohler Glaciers and the corresponding Crosson and Dotson Ice Shelves have undergone speedup, thinning, and rapid grounding-line retreat in recent years, leaving them in a state likely conducive to future retreat. We conducted a suite of numerical model simulations of these glaciers and compared the results to observations to determine the processes controlling their recent evolution. The model simulations indicate that the state of these glaciers in the 1990s was not inherently unstable, i.e. that small perturbations to the grounding line would not necessarily have caused the large retreat that has been observed. Instead, sustained, elevated melt at the grounding line was needed to cause the observed retreat. Weakening of the margins of Crosson Ice Shelf may have hastened the onset of grounding-line retreat but is unlikely to have initiated these rapid changes without an accompanying increase in melt. In the simulations that most closely match the observed thinning, speedup, and retreat, modeled grounding-line retreat and ice loss continue unabated throughout the 21$^{st}$ century, and subsequent retreat along Smith Glacier's trough appears likely. Given the rapid progression of grounding-line retreat in the model simulations, thinning associated with the retreat of Smith Glacier may reach the ice divide and undermine a portion of the Thwaites catchment as quickly as changes initiated at the Thwaites terminus.

## 1    Introduction

Glaciers along the Amundsen Sea Embayment (ASE) have long been thought to be vulnerable to catastrophic retreat (Hughes, 1981), and the major ice streams in the region have recently undergone significant speedup and grounding-line retreat (Mouginot et al., 2014; Rignot et al., 2014; Scheuchl et al., 2016). Largely due to synchronicity between variability in ocean temperature and glacier response, ocean-induced melting is thought to be the primary driver of these changes (Jenkins et al., 2010; Joughin et al., 2012). Oceanographic observations (Assmann et al., 2013) and modeling (Thoma et al., 2008) indicate that variable transport of warm circumpolar deep water (CDW) onto the continental shelf has caused substantial variability in sub-shelf melt over the past two decades, with melt thought to have temporarily peaked in the early 2010s (Jenkins et al., 2018). Melt rates influence the large-scale flow of ice streams by affecting ice-shelf thickness; thinner ice shelves provide less buttressing to ice upstream, and ice is forced to flow faster to increase strain-rate dependent stresses in the ice. Ice-flow modeling (e.g., Joughin et al., 2014) and glaciological observations (e.g., Rignot et al., 2014) suggest that the retreat of Thwaites and perhaps Pine Island

Glacier, the largest glaciers along the ASE, will continue under all realistic melt scenarios (Favier et al., 2014; Joughin
et al., 2010).

Despite their lower ice discharge relative to Thwaites and Pine Island Glaciers, Smith, Pope and Kohler Glaciers (see
Figure 1 for an overview of the area) have gained attention as some of the most rapidly changing outlets along the
Amundsen Sea Embayment (Mouginot et al., 2014). These glaciers, and the Crosson and Dotson Ice Shelves
downstream, have undergone >30 km of grounding-line retreat in recent decades (Rignot et al., 2014; Scheuchl et al.,
2016), leaving their grounding lines positioned more than 1 km below sea level, where they are vulnerable to warm
ocean waters (Jenkins et al., 2018; Thoma et al., 2008). By contrast, the Thwaites grounding line sits approximately
50 km downstream of the deepest portions of its basin (Rignot et al., 2014) and the Pine Island grounding line has
held a steady position on the retrograde slope at the seaward end of its overdeepening from 2009-2015 (Joughin et al.,
2016). Thus, the positioning of Smith Glacier's grounding line in the deep portion of its trough suggests that it is in a
more advanced stage of retreat than its larger neighbors. Indeed, Smith Glacier comprises one of the most extensive
instances of modern glacier retreat and can serve as an important example of a marine ice-sheet basin in an advanced
state of collapse.

Modeling of the grounded portion of the Smith, Pope, Kohler catchment indicates that these glaciers are committed
to further retreat on decadal timescales (Goldberg et al., 2015). However, this modeling was focused on transient
calibration and did not assess causes of retreat or examine likely changes over periods longer than 30 years. Additional
modeling work shows that the ice-shelf response is highly sensitive to the sub-shelf melt rates, which, when
determined from an ocean model, are in turn highly dependent on how well the bathymetry is resolved (Goldberg et
al., 2019). Regardless of the initial cause of retreat, the ice shelves are unsustainable at present melt rates, and Dotson
Ice Shelf may melt through in the next 50 years (Gourmelen et al., 2017). The ice presently within the Smith, Pope,
Kohler drainage could raise global mean sea level by a relatively modest 6 cm (Fretwell et al., 2012), but thinning can
lead to drainage capture and therefore increased loss of ice volume. Thus, due to a shared divide, rapid thinning could
potentially hasten the collapse of the larger reservoir of ice in the neighboring Thwaites catchment.

Although there is evidence of increased transport of warm ocean waters beneath these ice shelves, the complex nature
of ice-sheet dynamics involves the responses to past and present forcing. Present observations represent a combination
of adjustment to past imbalance and response to recent melt (e.g., Jenkins et al., 2018). In the case of Smith, Pope,
and Kohler Glaciers, multiple lines of evidence suggest that retreat began before widespread satellite observations
were first acquired (Gourmelen et al., 2017; Konrad et al., 2017; Lilien et al., 2018), though the exact cause and timing
of retreat initiation are unknown. Separating the effects of different forcings is key to understanding the extent to
which continued forcing is required to sustain retreat. Since future forcing is uncertain, identifying whether retreat is
inevitable within the expected range of ocean warming is particularly valuable. Because of the short length of the
satellite record, separating the compounded influence of the possible drivers of retreat is difficult with observations
alone, and numerical ice-flow models are an important tool for identifying plausible scenarios that could have resulted
in the observed changes to ice thickness, velocity, and grounding-line position.

Here, we describe a suite of model simulations designed to investigate which processes control the ongoing retreat of
Smith, Pope, and Kohler Glaciers. Our modeling experiments tested the effects of melt distribution, melt intensity,
basal resistance, and marginal buttressing on speedup, thinning, and grounding-line position. We compared these
modeled changes to remotely sensed observations in order to determine which processes have driven retreat over the
last two decades. After comparing the modeled velocity, surface elevation, and grounding-line position to
observations, we ran a subset of the simulations for a longer duration to investigate the sensitivity of the future
evolution of this system to a range of forcing.

Simulations of Antarctic ice streams generally require a melt forcing to determine the mass balance of the bottom of
the ice shelves. Spatially well-resolved sub-shelf melt rates have only recently been measured for ice shelves in the
ASE (Gourmelen et al., 2017; Shean et al., 2017), and these observations are limited by their brief record and low
temporal resolution. Thus, use of these high-resolution melt rates as inputs to prognostic ice-flow models that extend
further into the past or into the future requires extrapolation. To avoid such extrapolation, models are usually forced
with simple, often solely depth-dependent, parameterizations of melt (e.g., Favier et al., 2014; Joughin et al., 2010).
Significant progress has been made in coupling state-of-the-art ice and ocean models (e.g., De Rydt and
Gudmundsson, 2016; Jordan et al., 2018), though to our knowledge only one study has applied a fully coupled model
with moving grounding line to the geometry of a real glacier (Seroussi et al., 2017). Coupled simulations capture
spatial and temporal variability in melt rates but require substantial high-performance computing resources. Moreover,
modeled sub-shelf melt rates are highly sensitive to the sub-shelf bathymetry (Goldberg et al., 2019), which is difficult
to measure or infer due to the ice and ocean cover. Because these coupled ice-ocean models require additional
development and substantial high-performance computing resources, and are sensitive to uncertain bathymetry, they
are not yet readily available for assessing sensitivity to a suite of forcings.

While ocean forcing is thought to be the primary driver of retreat along the ASE, a glacier's sensitivity to sub-shelf
melt is modulated by additional processes. Grounding-line retreat exposes additional and, for a retrograde bed, deeper
sub-shelf area to melt, potentially increasing the integrated melt rate without any change in ocean heat content (De
Rydt et al., 2014). Additionally, ungrounding on a retrograde bed causes ice-flow speeds to increase due to the
nonlinear dependence of ice velocity on ice thickness. These feedbacks cause some grounding-line positions to be
inherently unstable, such that upstream perturbations to those grounding-line positions can lead to self-sustaining
retreat (e.g., Schoof, 2007). Changes to the effective viscosity of ice shelves, such as weakening from mechanical
damage, fabric development, or higher ice temperatures, can reduce the shelf's ability to transmit stresses and thus
reduce buttressing in the same manner as a decrease in the shelf's cross sectional area (e.g., Borstad et al., 2016).
Observations (Macgregor et al., 2012) and inverse modeling (Lilien et al., 2018) suggest that changes to viscosity
have indeed played a role in the speedup of Crosson Ice Shelf, and model sensitivity studies suggests that weakening
of several key regions of the ice shelves, particularly the shear margins or near the grounding line, would significantly
alter ice discharge (Goldberg et al., 2016, 2019). While some other processes, such as loss of terminal buttressing due
to retreat of the neighboring Haynes Glacier, may have destabilized Crosson Ice Shelf, changes to melt, marginal
weakening, and feedbacks between ungrounding and increased ice-flow speeds represent the most likely drivers of
retreat in this system.

## 2   Methods

We conducted a suite of prognostic numerical model simulations of Smith, Pope, and Kohler Glaciers, primarily using
a shallow-shelf (SSA) model implemented in the finite element software package Elmer/Ice (Gagliardini et al., 2013;
Zwinger et al., 2007). The shallow-shelf equations describe ice flow in two dimensions under the assumptions that the
ice is thin relative to its extent and that ice velocity is uniform with depth (i.e., the model is depth-averaged); while a
simplification, these assumptions are generally applicable to ice streams (MacAyeal, 1989) and have been applied to
other glaciers in the ASE (e.g., Favier et al., 2014; Joughin et al., 2014). To validate the use of these simplified ice
physics, we performed one simulation using a state-of-the-art full-Stokes (FS) ice-flow model, also implemented in
Elmer/Ice. In slower flowing regions where our inversion results show that internal deformation comprises a
significant portion of motion, incorporating the variation of velocity with depth may be important. The full-Stokes
simulation allows us to identify potential drawbacks of applying the simplified shallow-shelf model to this particular
system of glaciers.

### 2.1 Model setup

The model domain extended from the ice divide (determined from the measured velocity field) to the seaward edge
of the ice shelves' embayment (see Figure 1 for the extent of the domain). For all simulations, the horizontal mesh
resolution was 300 m near the grounding line and 3 km elsewhere. The full-Stokes domain was extruded to 9 vertical
layers, with 5 layers concentrated in the bottom third of the ice, giving an effective resolution of 20 to 500 m depending
on ice thickness and depth within the ice column. This resolution is generally considered sufficient to accurately
capture grounding-line dynamics (Pattyn et al., 2013), and sensitivity to mesh resolution is explored further in the
supplementary materials. The upper ice surface at initialization was found by adjusting a high-quality reference digital
elevation model (DEM) mosaic, derived from WorldView/GeoEye stereo imagery, to match expected conditions in
1996. This adjustment used thinning rates found from ICESat-1, the Airborne Topographic Mapper from NASA's
Operation IceBridge, and WorldView/GeoEye stereo DEMs (further description of the determination of this surface
can be found in Lilien et al., 2018). The bed elevations were determined from all publicly available airborne radio
echo sounding data, anisotropically interpolated to 1-km posting so as to weight measurements along flow more
heavily than those across flow; details can be found in Medley et al. (2014) and the supplementary materials to Joughin
et al. (2014). The advantage to this method of interpolation is that it is free of assumptions related to a particular state
of mass balance, unlike mass-conservation methods. The lower ice surface was then determined using the bed
elevations beneath grounded ice and using an assumption of hydrostatic equilibrium downstream of the 1996
grounding line. Firn-air content for the hydrostatic calculation was found by comparing coincident ice-thickness and
surface-elevation measurements over the ice shelves (supplementary materials of Lilien et al., 2018).

All model simulations were initialized to best match the transient state of these ice streams in 1996, the earliest year
with relatively complete maps of ice velocity in this area. The velocity measurements were acquired by
the European Remote-Sensing Satellites (ERS-1 and 2) and processed using a combination of interferometry and
speckle tracking (Joughin, 2002). Model initialization consisted of an iterative process using a full-Stokes,
diagnostic thermomechanical model in Elmer/Ice. We iterated between updating the temperature field and using
inverse procedures to infer the basal shear stress of grounded ice and the enhancement factors over floating ice.
These inferred fields minimized the misfit between modeled velocity and the measurements from 1996 (this initial
set of inversions is also described in Lilien et. al, 2018, where plots of the inferred enhancement are shown). In order
to minimize transient effects of data errors while capturing the real transient state of these ice streams in 1996, the
model was briefly relaxed by running forward in time for one year under constant forcing and with the grounding-
line fixed in place. Then, the inversions were repeated to infer the final inputs for the forward model. Further details
of inversion procedures, temperature initialization, and relaxation are provided in supplementary materials.

While it is difficult to assess whether the model accurately represents the true temperature, enhancement, and basal
slipperiness fields, modeled thinning rates at the end of relaxation give an indication of model self-consistency.
Conversely, the total change in surface height during relaxation gives a misfit between the model and available data
(though in part that relaxation may be compensating for errors in the data). Here, relaxation resulted in local changes
of up to 100 m near Kohler Glacier's grounding line and changes of at most 50 m elsewhere. While most of the
change during relaxation can potentially be attributed to errors in ice thickness caused by uncertainty in the bed
elevation, the large change on Kohler likely indicates that the surface elevations were also incorrect in that area.
Because determining the surface elevations at initialization required some extrapolation using longer term thinning
rates (see Lilien et al., 2018 for details), this misfit is not surprising and may reflect a change in the spatial pattern of
thinning during 1996-2003. At the end of the relaxation, thickness change rates were reduced to <10 m a$^{-1}$, which is
smaller than the observed rate of thickness change, except on Kohler Glacier where ~30 m a$^{-1}$ of thickening
persisted. While this is still a large rate of elevation change on Kohler, we were forced to choose between accepting
Kohler's unrealistic imbalance and possibly relaxing away the real imbalance on Smith and Pope. The potential
effects of the resultant transients upon the modeled retreat of Kohler are revisited in Section 4.3.1.
**2.2 Prognostic simulations**
We ran a suite of more than 20 ice-flow model simulations for at least 23 years, all beginning in model-year 1996.
These relatively brief simulations enabled comparison with observations, and 6 of these simulations were subsequently
run over 100 years to investigate the future evolution of these glaciers. Those 6 simulations were selected after the
full suite of shorter runs and were chosen to represent a range of retreat rates, some realistic and some slower than
observed, but all using realistic melt rates. Table 1 summarizes the inputs for all model runs, indicating the model

physics, run length, melt distribution and intensity, and any other forcing as described below. In all model simulations, time stepping used a backwards-difference formula with timestep size of 0.05 years. Calving was not explicitly modeled, but instead ocean pressure was applied on the downstream boundary at the mouth of the ice shelves' embayments where ice is allowed to flow out. This boundary condition would remain accurate for an advance since ice tongues extending beyond embayment walls do not provide additional back stress, but, if substantial ice loss caused the calving front to retreat behind the embayment walls, it could potentially result in an underestimate of ice loss during retreat.

Most of the model simulations used a Coulomb-type sliding law proposed by Schoof (2005) and Gagliardini et al. (2007), which takes the form

$$\tau_b = CN\left(\frac{\chi u_b^{-m}}{1+\chi}\right)^{\frac{1}{m}} u_b \qquad\qquad \text{Equation 1}$$

where $\tau_b$ is the basal shear stress, $u_b$ the basal velocity, $N$ the effective pressure, $C$ proportional to the maximum bed slope, $m$ the sliding law exponent, and $\chi = \frac{u_b}{C^m N^m A_c}$, where $A_c$ is a coefficient that is determined using the inversion results. This sliding law was derived to represent sliding over a rigid bed with cavitation behind obstacles, but its high- and low-pressure limits make it suitable for describing Antarctic ice streams. At high effective pressure, generally found in slow-flowing regions that may be underlain by hard beds, the sliding law approximates Weertman (1957) sliding ($\tau_b \propto u_b^m$). At low effective pressures, this Equation 1 approaches Coulomb-type sliding ($\tau_b \propto CN$), which is thought to be appropriate for sliding over soft beds (e.g., Iverson et al., 1998; Tulaczyk et al., 2000) and hard beds where fast-sliding with cavitation takes place (Schoof, 2005). We take $m = 3$, and assume that the effective pressure is equal to the ice overburden minus the hydrostatic pressure. With this assumption, Coulomb-like behavior only occurs within several kilometers of the grounding line, with Weertman-like behavior farther inland (Joughin et al., 2019). This assumption is valid for infinite hydraulic conductivity, but realistic, finite hydraulic conductivity would cause higher water pressures inland, which would lead to this parameterization underestimating the extent of Coulomb-like behavior. However, this assumption is often employed (e.g., Morlighem et al., 2010), and because coupling to a hydrologic model is beyond the scope of this study, we retain the assumption here. To some extent, errors in the assumption compensated for in the solution for the sliding coefficient, $C$, though it may introduce errors as the basal shear stress is reduced too drastically in response to inland thinning. For comparison, we ran four additional simulations with a commonly used Weertman-type sliding law ($\tau_b = A_w u_b^m$), with $A_w$ calculated from the same inversion results, again with $m = 3$.

### 2.2.1 Melt sensitivity experiments

We explored the effect of a variety of plausible melt forcings on the evolution of Smith, Pope, and Kohler Glaciers. The forcings can be separated into melt intensity (i.e. shelf-integrated melt) and its spatial distribution; simulations were conducted varying the melt intensity and distribution independently to determine their relative importance in

controlling retreat. Because of their low computational expense, we used simple prescriptions of melt: three depth-dependent parameterizations (Favier et al., 2014; Joughin et al., 2010; Shean, 2016), all tuned to fit the melt-depth relationship of nearby Pine Island Glacier, and an interpolation from previously published high-resolution melt-rate estimates inferred from Cryosat-2 by assuming hydrostatic equilibrium (Gourmelen et al., 2017), which was extended to cover both ice shelves. Hereafter, we refer to these melt distributions as F2014, J2010, S2016, and Cryo2, respectively. The parameterizations are intended to span a reasonable range of likely melt distributions, and none of them were expected to match the Cryosat-inferred pattern of melt exactly. Any depth-dependent parameterization will fail to span the range of melt rates observed at a given depth. However, the depth-dependent parameterizations capture the general form of the Cryo2-inferred melt rates, despite not having been tuned to Crosson and Dotson Ice Shelves (Figure 2).

Melt rates inferred from Cryosat 2 are limited to areas that were floating during the period of 2010-2016, which potentially complicates forcing the model with the Cryo2 distribution. If additional area beyond what was afloat in 2016 were to unground in a model simulation, some extrapolation would be needed to apply a melt forcing to that area. For the minor extrapolation that was necessitated by the retreat in these simulations, we first smoothed the melt rates to 2-km resolution then used nearest-neighbor interpolation to extend the rates inland. However, during the first 25 years of the model simulations, these extrapolated values were not required, so the limited extent of the inferred melt rates does not affect comparison between modeled and observed retreat.

During each timestep from model years 1996-2014, each melt distribution was re-scaled to match the time-varying shelf-total melt rate, as derived from flux divergence. Note that this scheme differs from prior studies (Favier et al., 2014; Joughin et al., 2010, 2014) that instead fix the parameterization for a particular run and accept the resulting temporal variation in melt rate as the depth of shelf's underside evolves. Comparative advantages and disadvantages of our approach are discussed in detail in section 4.4.2, but this choice was mainly made to limit melt rates to realistic values during the period with observations. The melt intensity was determined by linear interpolation between available measurements of shelf-total melt obtained from flux divergence measurements through time (Lilien et al., 2018). In general, this scheme requires adjusting the depth-dependent parameterizations down from the "1x" versions by a factor of 4-5; such scaling is unsurprising given the large differences between the Dotson and Crosson cavities and the Pine Island Glacier cavity for which the parameterizations were originally tuned. Though the Cryo2 rates agree to within errors with the flux divergence estimates over Dotson during 2010-2014 (Lilien et al., 2018), the forcing was scaled down by ~20% at the start of the simulations to match the relatively low melt rates in 1996. Through the simulations, the scaling factor for melt was generally increased to force the observed increases in melt. For the depth-dependent parameterizations, this increase was compounded by the need to compensate for the rapid decrease of ice-shelf draft due to intense melt at depth, which can result in the shelves "shallowing out" of high melt rates over most of their area. Thus, the scaling through time varied significantly based upon how quickly the ice-shelf draft shallowed and how much new area became exposed to the ocean and contributed to the shelf-total melt rate. After 2014, when melt-rate estimates are no longer available, the scaling was fixed to the value determined for 2014 and the

total melt rate was allowed to vary as in previous studies. For partially floating elements, melt was applied only over
the floating portion, and the model resolution employed avoided significant sensitivity to this choice (Seroussi and
Morlighem, 2018).

We also conducted simulations changing melt intensity to twice that observed (simulations 3, 11, 18, and 25 in Table
1). To vary the melt intensity, we again rescaled the parameterization at every timestep through 2014 in order to force
the total melt rate to match twice the observations. To distinguish these from what previous authors refer to as "1x",
"2x", and "4x", we instead refer to the different intensities as "1Obs" and "2Obs". It is important to note that in the
prior studies, "Nx" referred to scaling of the parameters, which, due to shallowing of the ice-shelf draft, could lead to
substantially less melt than N times the observations. Our scaling ensures that during the period of observations, 2Obs
actually doubled the shelf-wide integrated melt.

### 2.2.2 Marginal weakening experiments

We manually masked the areas within 10 km of the margins of Crosson and Dotson Ice Shelves and applied an ad-
hoc change to the depth-averaged enhancement factor over these areas to test the model's sensitivity to marginal
weakening. These runs were conducted using the shallow-shelf model and used an enhancement factor of 4 (a 44%
reduction in $B$) to weaken the margins. These weakening experiments were done with all four melt distributions at
1Obs melt intensity. One additional simulation was run with an enhancement factor of 1.8 (a 17% reduction in $B$)
using the J2010 melt parameterization at 1Obs intensity (simulation 5 in Table 1). In order to test the effect of marginal
weakening in the absence of any increase in melt, an additional set of simulations were conducted fixing the melt
parameterization to its 1996 scaling and applying the enhancement factor of 4; these simulations again used each of
the four melt distributions at 1Obs intensity (simulations 4, 12, 19, and 26 in Table 1). We refer to these experiments
with weakened margins but fixed melt parameterization as "control melt" simulations (simulations 6, 13, 20, and 27
in Table 1).

### 2.2.3 Forced ungrounding experiments

Since model simulations cannot be expected to perfectly replicate observed grounding-line retreat, we ran an
additional suite of experiments to test the effect of the ungrounding itself on thinning and speedup. These simulations
allow us to assess whether feedbacks between ungrounding, thinning, and speedup may have caused the observed
retreat, and to separate errors in modeled grounding-line retreat rates from their effects on ice-flow speed and thinning.
To estimate the grounding-line position at times between the three available measurements (1996, 2011, and 2014),
we linearly interpolated the time of ungrounding along a suite of flowlines spaced approximately every kilometer
across flow, creating maps of the grounded area every 0.1 years. At each model timestep through a forcing period
(1996-2001 or 1996-2014 depending on the simulation), the grounding-line position was set to match the nearest
grounding map, without changing the ice geometry, (i.e. the basal shear stress was set to zero and melt was applied
under ungrounded area). We only forced retreat and not the re-advance of Kohler between 2011 and 2014 since forcing
re-advance is complicated by the changing geometry after the ice goes afloat. After the period of forced ungrounding
finished, the grounding line was allowed to retreat freely based upon hydrostatic equilibrium. Simulations were
conducted with all four melt distributions at 1Obs intensity and with both 5 and 18 years of forced ungrounding
(simulations 7-8, 14-15, 21-22, 28-29 in Table 1).
**3    Results**
Model outputs are composed of the spatio-temporal evolution of a number of variables, notably ice velocity, ice
thickness, and grounding-line position. To distill this many-dimensional output into a manageable format, we focus
on comparing the changes to grounding-line position and ice-surface speeds along the centerlines of the three main
outlet glaciers under various forcings.
**3.1 Melt variability**
Figure 3 shows the results of the eight experiments designed to evaluate the melt intensity and distribution
(experiments 1-3, 10-11, 17-18, and 24-25 in Table 1). Collectively, the results show that grounding-line position and
the pattern of thinning are highly sensitive to the spatial distribution of melt. For the 1Obs experiments, there is <10
km of grounding-line retreat in the shallow-shelf simulations, and the retreat that does occur happens after model year
25. Amongst the 1Obs shallow-shelf simulations, only the one with J2010 melt shows more than 2 km of retreat,
during which time Smith Glacier's grounding line retreats by ~9 km. The full-Stokes simulation with 1Obs, however,
shows substantial (30 km) retreat along Smith Glacier during that time, in relatively good agreement with the
observations.

Over the first 25 years, retreat in the shallow-shelf models is generally confined to simulations with the 2Obs melt
forcing and is greatest with parameterizations that concentrate melt at depth. While the timing of retreat onset varies
with melt forcing, the 2Obs parameterizations generally yield similar retreat along Smith and Kohler glaciers (see
observed change from 1996 to 2011 in Figure 3f). An exception is the Cryo2 melt, which consistently produces the
least retreat. For Pope glacier with 2Obs forcing, the extent of the retreat varies greatly with melt distribution, ranging
from 0 to 18 km compared to the observed ~10-km retreat. Along Smith and Kohler Glaciers, simulations with the
J2010 distribution retreat most rapidly, followed by S2016, F2014, and Cryo2. Melt rates near the grounding line need
to reach some threshold before retreat commences; in the shallow-shelf model of Smith Glacier, retreat of the
grounding line does not begin unless melt rates of ~100 m a$^{-1}$ or higher are reached near the grounding line. Retreat
commences more easily in the full-Stokes model, requiring only ~50 m a$^{-1}$ of melt.  The grounding-line retreat rate of
Pope Glacier, which has a slightly shallower (~750 m.b.s.l.) grounding line, has a less direct relationship with melt
distribution. While retreat initiates most quickly with the J2010 parameterization, it is eventually overtaken by retreat
with the S2016 and F2014 parameterizations (Figure 3c).
**3.2 Marginal weakening**
We ran nine simulations with weakened margins, and all displayed notable differences in grounding-line position and
speedup compared to the simulations with no weakening. Figure 4 shows the effects of weakening on grounding-line

retreat and ice-flow speedup. The grounding-line positions of Smith and Pope Glaciers are sensitive to the shelf viscosity. With the J2010 melt parameterization, the retreat for Smith Glacier initiates ~10 years sooner with enhancement of 4 in the margins (Figure 4a-b). While this lag can lead to substantial differences in grounding-line position at any given time, the simulations with full-strength margins generally continue to retreat and reach that same state 10 years later. The notable exception is the simulation with the S2016 melt, which shows >10 km more grounding-line retreat when the margins are weakened (Figure 4a). Kohler Glacier's grounding line also retreats sooner with enhanced margins, but as retreat progresses grounding-line position does not differ by more than ~2 km from the unweakened case (Figure 4c). In the case of the S2016, F2014, and Cryo2 melt forcings, within 50 years, weakening of the margins causes grounding-line retreat on Pope and Kohler glaciers that did not take place even in 100 years without marginal weakening (Figure 4a and c). Simulations with enhancement of 1.8 display approximately half as much change in the timing of retreat as an enhancement of 4 does (not shown). Effects of marginal strength on ice speeds differ markedly between the two ice shelves; Crosson/ Pope flows almost 50% faster in some regions (Figure 4d-e) when the margins are weakened while Dotson/Kohler speeds are nearly insensitive to the strength of the margins (Figure 4f).

Although some of the simulations with weakened margins show more retreat, these simulations all are forced using the 1Obs melt intensity and thus incorporate the increases in melt observed between 1996 and 2014. In the "control melt" simulations with weakening but with the melt parameterization fixed at 1996 values, there is only minor grounding-line retreat over the 50-year duration of the simulations. If the weakening alone were sufficient to cause grounding-line retreat, we would expect to have seen retreat in these simulations.

### 3.3 Forced ungrounding

Figure 5 shows the results of the simulations in which the grounding line was forced to migrate at the rate observed. The forced ungrounding had differing effects depending on the melt distribution, and in some cases no subsequent grounding-line retreat ensued after the period of imposed ungrounding. In simulations with the 5-year forced ungrounding, the grounding line is able to stabilize temporarily (Figure 5a-c), though retreat subsequently ensues on Smith Glacier for the melt distributions that concentrate melt at depth (J2010, S2016). Ice-flow speeds on Pope and Kohler glaciers are relatively unaffected by the forced 5-year grounding-line retreat, but when forced through 2014 (18 years) they display some speedup as well (Figure 5d and f). In the case of Smith Glacier, the effect of exposing additional area to melt and decreasing basal resistance results in substantial speedup near the grounding line that continued over 25 years following the period of forced ungrounding. For the 18-year forced-ungrounding simulations, little grounding-line retreat occurs on any of the glaciers in the subsequent 25 years, leaving the grounding lines within 5 km of their 2014 positions.

### 3.4 Longer term simulations

Figure 6 shows the evolution of the ice volume and grounding-line position for the centennial-scale simulations, displaying sustained loss of ice volume through 2100 CE. These six simulations (simulations 2, 4, 10, 17, 19, 24 in

Table 1) were simply extensions of model runs mentioned above, chosen to represent a range of retreat scenarios with
realistic melt intensity; four used the different melt distributions at 1Obs intensity and no marginal weakening while
two used the J2010 and S2016 melt distribution at 1Obs intensity with marginal weakening. Simulations with marginal
weakening and/or the J2010 melt parameterization show continuing grounding-line retreat throughout the simulation
(Figure 6c-d). In simulations with substantial (>50 km) of retreat, the grounding line of Smith Glacier eventually
extends upstream of Kohler, and the grounding lines of these two glaciers merge. Even in these simulations with the
most retreat, melt rates remain below 75 Gt a$^{-1}$ (within 25% of 2014 levels) for most of the 21$^{st}$ century before gradually
increasing to 120 Gt a$^{-1}$ between 2080 and 2100, as more deep ice is exposed to melt. With these relatively modest
melt rates, the overall contribution to sea-level rise still ranges from 6–to–10 mm by 2100 and Smith Glacier's
grounding line retreats by >80 km in the simulations with J2010 melt distribution. Despite continued loss of ice
volume, substantial grounding-line retreat never initiates when using the F2014, S2016, or Cryo2 melt distributions
with 1Obs intensity. Even in these simulations with little retreat, contributions to sea level exceed 2 mm by 2100.
**4    Discussion**
We first evaluate how different parameter choices in the model affect its ability to reproduce the extensive grounding-
line retreat between 1996 and 2014, then consider the implications for the future retreat of the system. We then discuss
how model simulations compare to the observations of ice-flow speed and thinning from 1996-2018 and evaluate how
necessarily subjective modeling choices may have affected these results.
**4.1 Conditions needed to match observed grounding-line retreat**
The extensive, observed 30-km retreat (Rignot et al., 2014; Scheuchl et al., 2016) provides a simple metric for whether
model simulations match the data. Along the Smith centerline, the bed depth remains at around 1 km b.s.l. for the first
10 km upstream from the grounding line before deepening to close to 2 km b.s.l. over the following ~10 km (Figure
3f), and in many simulations the grounding line never retreats off this relatively flat, shallow portion of the bed. This
geometry leads to an essentially bimodal distribution of grounding-line position along the Smith glacier centerline.
Model simulations where the retreat reaches the retrograde slope past 10 km all reach >20 km of grounding-line retreat
(Figure 3c). While only the full-Stokes simulation matches the timing of the observed retreat under 1Obs melt
intensity, the stepped pattern of retreat is similar regardless of model physics (discussed more in Section 4.4.1). We
partition the simulations into those that display 15 km or more grounding-line retreat on Smith glacier, regardless of
the timing, and those that do not; those that display this large retreat are considered generally good matches to the
observed grounding-line positions where ~30-km of retreat was observed. The simulations that matched this large
retreat were: those with the J2010 melt parameterization with 1Obs or 2Obs melt, irrespective of marginal weakening;
those with the S2016 or F2014 parameterization and 2Obs melt; and the simulation with the S2016 parameterization,
1Obs melt, and marginal enhancement of 4 (Table 1).

We find that grounding-line position is controlled by a combination of melt distribution, melt intensity, and marginal
weakening, though melt near the grounding line (a product of melt distribution and intensity) is the primary driver of
retreat. This result confirms the conclusion of previous work that has also highlighted the importance of the melt
distribution for determining ice-shelf stability (Gagliardini et al., 2010; Goldberg et al., 2019; Seroussi and Morlighem,
2018). To match the observed grounding-line retreat using realistic (1Obs) melt intensity, the models suggest that melt
must have been concentrated near the grounding line. Concentrated melt at depth is expected given that the warm
CDW which drives melt generally intrudes at depth (e.g., Jacobs et al., 2012). However, without elevated melt
intensity (relative to 1996) or greater concentration of melt at the grounding line than considered by our melt forcings,
the "control melt" simulations show that the modeled grounding-line positions of Smith, Pope, and Kohler glaciers
would have remained stable for the 50 years following 1996.

The stable grounding-line position found by forcing the model with Cryo2 melt (Gourmelen et al., 2017) may result
from underestimation of melt near the grounding line in 1996, either due to the difficulty of using satellite altimetry
to infer melt rates in an area not in hydrostatic equilibrium (Fricker and Padman, 2006; Rignot, 1998) or due to a
change in distribution of melt between 1996 and 2010. Since melt rates were inferred over 2010-2016, if melt were
highest near the grounding line in 1996 but subsequently the area of peak melt moved upstream, the 2010-2016 rates
may be much lower those in 1996 near the grounding line at that time. This mismatch in observation time and model
forcing could have then resulted in the model never beginning to retreat into areas of concentrated melt. Moreover,
even once retreat was triggered, the inferred melt rate beneath areas that ungrounded during 2010-2016 mixes periods
of no melt and more intense melt, thus causing underestimation of the annual-average melt during the periods when
the ice was ungrounded. While our rescaling of parameterizations can increase the melt rates at the grounding line as
the shelf-averaged ice draft decreases, the Cryo2 distribution does not allow the shelf to shallow out of melt, and so
any underestimation of melt near the grounding line persists through the simulation. Thus, effective melt rates at the
grounding line are lowest using the Cryo2 distribution, and they remain too low to induce retreat. Estimates of melt
rates from ocean models should eventually provide a better option for forcing models, but computational constraints
and poorly constrained cavity geometry prevent their widespread application at present (e.g., De Rydt and
Gudmundsson, 2016).

It is possible that weakening of the margins of Crosson affected the timing of grounding-line retreat. Our model
simulations applied an ad-hoc enhancement of 4 to the margins, which is akin to ~5° C of warming (Cuffey and
Paterson, 2010), development of a relatively weak anisotropic fabric (Ma et al., 2010), or damage due to rifting (e.g.,
Borstad et al., 2013). While snapshot inversions for ice-shelf viscosity in 1996, 2011, and 2014 indicate some
weakening of Crosson Ice Shelf (Lilien et al., 2018), this weakening cannot be definitively identified as having been
caused by a particular process (e.g. loss of a pinning point or rifting). Thus, we are unable to identify if the weakening
of the margins was triggered by grounding-line retreat itself or was externally triggered and helped initiate grounding-
line retreat. We consider it unlikely, regardless of their cause, that changes to the strength of the shelf were the primary
cause of retreat since the simulations with marginal weakening but no increase beyond 1996 melt rates showed little
retreat. Additionally, inversion results do not show significant weakening of Dotson Ice Shelf through this time (Lilien
et al., 2018), suggesting that weakening was not the cause of Kohler Glacier's retreat even if it affected Pope and
Smith Glacier, and thus does not explain widespread retreat in the area.

The modeled grounding-line positions demonstrate the stepwise nature of grounding-line retreat and highlight the
complexity of assessing whether unstable retreat is taking place. Previous modeling has found that grounding lines
tend to remain in relatively favorable positions for a period before abruptly retreating (e.g., Joughin et al., 2010), and
the presence of grounding-line wedges at various points on the continental shelf indicate that retreat since the last
glacial maximum followed a similar stepwise pattern with extended periods of stability (Graham et al., 2010; Smith
et al., 2014). Similarly, exposure dating of glacial erratics along Pine Island Glacier indicate that during the Holocene
it experienced long periods of slow retreat punctuated by decades or centuries of rapid thinning (Johnson et al., 2014).
Our forced ungrounding experiments were designed to test whether the grounding line was situated such that some
perturbation necessarily led to a continued step back to a new stable grounding-line position. While forced
ungrounding for 5 years resulted in retreat of one simulation that otherwise remained stable (17 vs. 21 in Table 1),
even with elevated melt intensity the grounding line was able to stabilize on the retrograde slopes under some melt
distributions (Figure 5), at least over the period of our simulations. Additionally, regardless of melt distribution, little
further retreat was found in the 25 years following 18 years of forced ungrounding (Figure 5). The re-stabilization of
the retreated grounding line indicates that small perturbations do not necessarily lead to immediate retreat, although
25- to 50-year simulations may simply be too short to capture the retreat that may eventually ensue. These forced
ungrounding experiments also serve as a check upon the low temporal resolution of the melt forcing; the shelf-total
melt was linearly interpolated between measurements in 1996 and 2006, and a brief period of elevated melt could
have perturbed the grounding line during a subset of that time. However, the simulations with 5 years of forced
ungrounding suggest that such a perturbation would not have led to immediate and sustained grounding-line retreat.
Rather, sustained high melt rates at the grounding line appear to be necessary to cause the continuing grounding-line
retreat that has been observed.

## 4.2 Centennial simulations

The centennial-scale simulations can be broadly categorized as those that emulate observed grounding-line retreat (i.e.
display more than 35 km of retreat) and those that retreat less than observations. Those simulations that emulate retreat
(2, 4, and 19 in Table 1) all continue to produce retreat into the future. Even those simulations that do not capture the
magnitude of recent retreat yield continuing mass loss resulting in over 2 mm of contribution to global mean sea level
by 2100 (Figure 6). In the simulations with the 1Obs J2010 melt parameterization, nominally equivalent to no increase
beyond 2014 melt forcing, ice losses exceed 8 mm sea-level equivalent and reaches 10 mm when marginal weakening
is included. With the S2016 parameterization and marginal weakening, the grounding line also continues to retreat,
albeit at a more moderate pace, and losses still reach 6 mm sea-level equivalent by 2100. This simulation with the
S2016 forcing and marginal weakening is essentially a minimum loss scenario amongst simulations capable of
producing the observed retreat; shelf-total melt rates after 2014 remain below 50 Gt $a^{-1}$, lower than observed in 2006-
2014, yet grounding line retreat and sea-level contribution continue unabated. Thus, these three simulations suggest
that these glaciers will likely contribute 6 mm of sea-level rise over the coming century, even if shelf-integrated melt
rates remain at about their levels in recent years. Moreover, the delayed grounding-line retreat compared to
observations suggests that these projections are more likely to underestimate than overestimate future ice loss. Given
the retreat produced by the simulations with the lowest melt, and grounding-line retreat rates suggest that these
simulations underestimate loss, it is unclear whether Smith Glacier could now reach a new stable configuration before
the grounding line recedes to the head of its trough.

While the volume above floatation in the Smith, Pope, Kohler catchment is modest compared to some Antarctic
catchments, if thinning were to extend to the divide with the Thwaites catchment, additional losses could result. Due
to the extensive grounding-line retreat already undergone by Smith Glacier, the simulations with the J2010 melt
distribution suggest that substantial (>50 m) thinning could reach the divide shared with Thwaites by the end of the
21$^{st}$ century. This thinning could further contribute to the destabilization of the interior of Thwaites caused by changes
at Thwaites' terminus (Joughin et al., 2014). Because of their limited domain, our model simulations are unable to
assess the effects of divide migration on regional ice loss, and bed topography might isolate the loss to Smith's present
catchment. However, given the potential for divide migration, studies concerned with the stability of Thwaites Glacier
on timescales longer than ~100 years may underestimate ice loss if they do not account for potential drainage capture
by Smith Glacier.
**4.3 Comparison to other observations**
Here we assess how different model forcings affect the match between the simulations and the observations of thinning
and speed change. For all simulations, there are substantial differences between modeled and observed ice-flow speeds
and thinning rates, which need to be assessed carefully in order to understand the limitations of the model results. By
evaluating this mismatch, we can identify the direction in which the model simulations likely err and work to identify
processes that may be important for these glaciers but are not captured by our modeling results.
**4.3.1 Ice surface elevation**
In Figure 7, we compare modeled and measured ice-surface lowering. The comparison is confined to ice that was
grounded in 1996 since observations have greater signal-to-noise ratio over grounded ice; on grounded ice, all thinning
is expressed as surface lowering whereas on floating ice only ~10% of thinning is expressed at the surface.
Observations of surface lowering were derived from the various altimetry products described in section 2.1. The full-
Stokes simulation slightly overestimates thinning along Smith Glacier while producing thickening upstream of Kohler
Glacier's grounding line (Figure 7a). In general, the shallow-shelf simulations approximately match the pattern of
observed surface change downstream of the grounding line, but show too little thinning upstream (Figure 7b-d). Even
the simulations with 1Obs forcing that showed the most thinning slightly underestimate surface lowering. Part of this
difference may reflect errors in the bed elevation; if the true bed elevation were greater than estimated in the bed
product we used, a larger portion of dynamic thinning would have directly affected the surface height rather than
contributing to ice-draft shallowing. An additional portion of the model-data mismatch is likely due to timing of
retreat; a delayed response of the model could lead to underprediction of surface lowering. Given that the shallow-
shelf simulations have delayed grounding-line retreat, it is unsurprising that they generally underestimate surface
change.

The thickening (or lack of thinning) on Kohler may result from difficulties in initiating a model of an out-of-balance
system. Melt and calving in 1996 were already larger than accumulation, likely due to elevated melt on Kohler (Lilien
et al., 2018), and it is possible that the relaxation of the model prior to the simulations dampened real surface changes
rather than artifacts from data errors in the Kohler drainage. Regardless of its cause, this discrepancy is transient and
surface lowering eventually propagates up the trunk of Kohler as in observations. However, this thickening on Kohler,
along with the shallow-shelf simulations' delayed grounding-line retreat and thinning, suggest that the simulations
may underestimate future ice loss.
**4.3.2 Ice-flow speed**
We compare the model results to velocity mosaics for 2006-2012, 2014, and 2016-2018. The 2007-2010 velocities
are derived from the Advanced Land Observation Satellite, processed using a combination of interferometry and
speckle tracking (Joughin, 2002). We used feature tracking of Landsat-8 imagery to obtain velocities for the 2014–
2015 austral summer. Velocity data for 2006 and 2011 are part of the NASA MEaSUREs dataset (Mouginot et al.,
2014). We determined the 2016–2018 velocity using speckle-tracking applied to data from Copernicus Sentinel-1A/B
data. These observations indicate speedup both near the grounding lines of Smith and Kohler Glaciers and farther out
on Crosson Ice Shelf (Mouginot et al., 2014). While the speedup near the grounding line is likely due a loss of basal
resistance as a result of ungrounding, the speedup of the outer shelf may be due to changes in shelf viscosity or loss
of buttressing at the outer right corner of the ice shelf due to the breakup of the Haynes glacier tongue (Lilien et al.,

514    2018).


The simulations indicate that ungrounding primarily affects speeds near the grounding line while speeds farther out
on the shelf remain constant or decrease (Figure 5). This heterogeneity results from buttressing; if the shelves were
spreading freely, a change in grounding-line speed would cause an equal change in the speed of the shelves.
Conversely, speedup of the outer portion of the ice shelves is likely a result of local changes to buttressing since
speedup is not observed in the region immediately upstream. The model experiments with weakened margins find
speedup along the Pope Glacier centerline on the outer portion of Crosson Ice Shelf (Figure 4d). While the modeled
speed changes in the simulations with weakening closely match the observed speeds 40-60 km from the 1996 calving
front, they show too little speedup closer to the front. This discrepancy along the shelf suggests that part of the observed
changes in speed may be a result of forcing near the calving front, possibly associated with a loss of buttressing due
to the breakup of the Haynes glacier tongue around 2002 or the progressive rifting of this area. While the simulations
with weakened margins do not fully capture the observed velocity changes near the shelf margin, the marginal
weakening does cause the model to more accurately reproduce speedup of the bulk of Crosson Ice Shelf. There are a
variety of possible reasons that the model does not capture the full spatial complexity of the observed speedup, for
example weakening of the ice shelves, bed elevation errors, or inferred basal resistance being too low, and we cannot
identify a single cause.

For the grounded ice, the simulations tend to under predict speedup on Smith Glacier, while generally overpredicting
speed changes on Kohler Glacier. The timing of the speedup corresponds with the timing of rapid grounding-line
retreat, so the delay in modeled grounding-line retreat likely causes the delay in modeled speedup. The scarcity of
observations of grounding-line position and ice velocity earlier in the satellite records complicates the interpretation.
Reliable grounding-line positions are unavailable between 1996 and 2011, and ice velocities are unavailable between
1996 and 2006. Substantial retreat occurred during this time period, and transient speedup could have occurred during
the gap in the observations.

## 4.4 Model limitations
We now evaluate effects that our choices in model complexity and melt forcing have on interpreting our results. In
addition, the relative insensitivity of the modeled retreat to our choice of sliding law and of the model resolution are
shown in supplementary materials.
### 4.4.1 Model complexity
Full-Stokes models require significantly greater computing resources than shallow-shelf models of similar resolution.
In the case of our simulations, the shallow-shelf simulations took ~1% of the CPU hours of an equivalent full-Stokes
simulation, allowing the use of local workstations rather than high-performance computing resources. Thus, using the
simplified physics of shallow-shelf models is desirable in cases where it is sufficient to capture the relevant processes.
While we find slower initiation of retreat with shallow-shelf than with full-Stokes models, after initialization the
pattern of retreat is similar between both classes of models.

Uncertainties in the model inputs, and necessary choices when initializing models, create significant spread in model
retreat rates that could explain the difference between full-Stokes and shallow-shelf simulations. For example, at Pine
Island Glacier, uncertainty in bed elevation propagates to uncertainty in the timing of retreat of around ±5-10 years
depending on assumptions about the spectrum of the bed roughness (Sun et al., 2014). Moreover, with idealized
geometry, L1L2 models, a class of depth-integrated models with slightly greater complexity than shallow-shelf
models, are more sensitive to high-frequency noise than full-Stokes models (Sun et al., 2014), suggesting the
possibility that the uncertainty in bedrock elevation may affect the full-Stokes and shallow-shelf models in different
ways.

The spacing of bed elevation measurements in our study region does not resolve detail with wavelengths of ~5 km
and below. In addition, noise with longer wavelengths may be present if there are systematic biases in the
measurements. Without constraints on this roughness, we cannot realistically assess how bed uncertainty may have
affected the two types of models differently. However, comparison of observed and modeled grounding-line position
and surface elevation suggest that errors in the bed dataset have indeed affected our results. The path of ungrounding
of Smith Glacier for most model simulations progresses directly through an area that has been identified as having
remained grounded through 2014 (Rignot et al., 2014; Scheuchl et al., 2016) despite the thinning rates in that area
matching observations there. If the bed elevations were accurately captured by the bed product, accurately modeling
thinning would be sufficient to accurately model retreat. By contrast, in an area where the bed is shallower than the
bed product suggests, ungrounding would occur too early in the model and a greater portion of thinning would be
expressed as ice-draft shallowing rather than surface lowering. Since the model finds ungrounding of a portion of
Smith while approximately matching thinning rates there, it is likely that the bed is shallower there than the bed
product indicates. Thus, we have strong evidence that errors in the bed elevation may have changed the ungrounding
in our simulations, but we are unable to constrain the different ways this would have affected different simulations.

Limitations of the assumptions in the sliding law are another potential source of differences between the models.
Recent work shows that alternatively parameterized versions of Equation 1 (regularized Coulomb friction) extend
plastic behavior much farther inland to yield better agreement with observations on Pine Island Glacier (Joughin et
al., 2019). The friction law in Equation 1 relies on a height-above-flotation parameterization for effective pressure,
ignoring hydraulic gradients and limiting Coulomb (plastic) behavior to near the grounding line. Thus, the friction law
used here may cause initially slow retreat in the shallow-shelf model to result in persistent differences from the full-
Stokes model. Regularized Coulomb friction could potentially lead to faster modeled retreat rates in some simulations
as plastic behavior follows the grounding line inland, thereby improving model data agreement beyond that found
here.

Time to full relaxation in the model spin-up, differences in the inferred basal shear stress resulting from inversion
procedure implementation, or different response to errors in surface elevation all may explain an additional portion of
the difference between full-Stokes and shallow-shelf models. Assessing the effect of uncertainties in these parameters
would require considerable investigation that is beyond the scope of this study. However, given that there are known
errors in the bed topography, and that the unconstrained frequency of bed noise affects the models differently, it is
possible bed errors alone could change the timing of retreat by as much as the model-data mismatch. Thus, while the
difference in timing between full-Stokes and shallow-shelf models might indicate substantially better full-Stokes
performance for at least one of the three glaciers, it could also reflect the uncertainty and not indicate that one type of
model is better suited to describing this system. Indeed, while the full-Stokes model better matches the timing of
retreat on Smith Glacier, it finds thinning rates that are a poorer match to observations and does not do a better job
than the shallow-shelf model at reproducing retreat on Pope or Kohler glaciers. Unfortunately, we did not have the
computational resources for a suite of full-Stokes runs sufficient to make a robust comparison of relative performance.

### 4.4.2 Melt forcing scheme

The application of the melt parameterizations in this study differs from previous work because, at each timestep where there are data, it rescales the parameterization so that model matches the observed shelf-wide integrated melt through time (Lilien et al., 2018). The primary advantage of this scheme is that it prevents the large, likely unrealistic changes to the shelf-total melt rate that occur as concentrated melt at depth causes the ice-shelf draft to shallow. We utilized this scheme primarily out of necessity; the grounding lines of Smith and Kohler Glaciers are sufficiently deep that without scaling the melt forcing, the shelf-total melt rates are drastically out of balance as simulations begin, and substantial retreat ensues before the shelf is able to shallow out of the intense melt, thus leading to sustained, unphysically high melt rates. On the other hand, the continuous-rescaling scheme dampens feedbacks between the grounding-line retreat and the melt rate. Whereas a fixed parameterization generally causes an initial increase in shelf-total melt in response to a retreat of the grounding line since greater sub-shelf area is exposed, this continuous-scaling scheme will reduce the scaling of the melt distribution in response to that retreat. The continuous-rescaling scheme may thus unrealistically dampen feedbacks leading to rapid retreat, since increasing exposure of sub-shelf area may truly increase the total melt rate if there is sufficient heat content in the nearby ocean. Because melt is not solely a function of depth, any depth-dependent melt parameterization faces tradeoffs between fidelity to observations and simplicity, but the scheme used here is a reasonable compromise for a study that needs quasi-realistic melt rates at the beginning of simulations to enable comparison between model and observations.

## 5   Conclusions

Using reasonable melt intensity distributed with simple, depth-dependent parameterizations, our model simulations are able to reproduce the recent speedup, thinning, and retreat of Smith, Pope, and Kohler Glaciers, albeit with some uncertainty in the timing. These simulations suggest that in 1996 Smith Glacier was in a state of precarious stability, but nonetheless elevated melt rates were needed to cause the observed grounding-line retreat. Even when shelf-integrated melt rates were increased, modeled retreat only occurred when that melt was concentrated near the grounding line and not farther out on the shelf. Explicit forcing of some retreat was also insufficient to cause the extent of grounding-line retreat that has been observed, as the grounding line was able to re-stabilize, at least temporarily, unless the melt was concentrated at depth. While weakening of the margins of Crosson Ice Shelf may have played a role in the speedup of the shelf or in the timing of grounding line retreat, it is unlikely that such a change precipitated the observed changes. Comparison to observations indicates that our model simulations underpredict the speedup and thinning of these glaciers, but despite this underprediction those model simulations that successfully reproduce recent grounding-line retreat continue to show grounding-line retreat into the future. We find that the rate of grounded ice loss is likely to grow in the coming decades as retreat progresses. These simulations indicate that >6 mm of sea-level contribution is likely by 2100, even if the total melt remains around current levels. By the end of our ~100-year simulations, thinning has extended to the ice divide separating Smith and Kohler from Thwaites Glacier, indicating the potential for Smith's retreat to hasten the destabilization of that larger catchment.

**Acknowledgements**

Computing resources were provided by the NASA high performance computing center. DL was supported by NASA headquarters through the NASA Earth and Space Sciences Fellowship (NNX15AN53H). NASA (NNX17AG54G) also provided support for contributions by IJ and BS. NG was supported by European Space Agency contract CryoTop Evolution 4000116874/16/I-NB. We thank Martin Truffer and the anonymous reviewer for careful comments that helped improved the clarity of the manuscript.

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

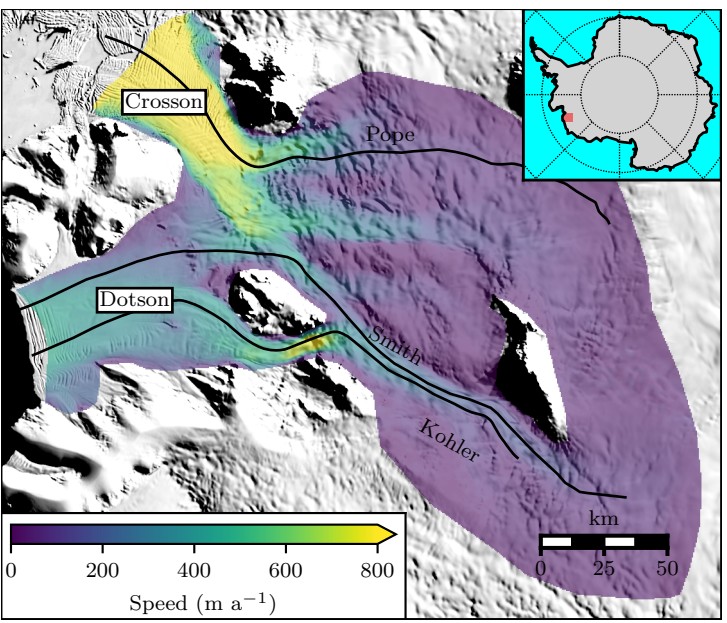


**Figure 1. Study area.** Colors show InSAR-derived velocities from 1996, plotted only over the area of the model domain. Black lines indicate flowlines for the three outlet glaciers. Inset shows location of study area.


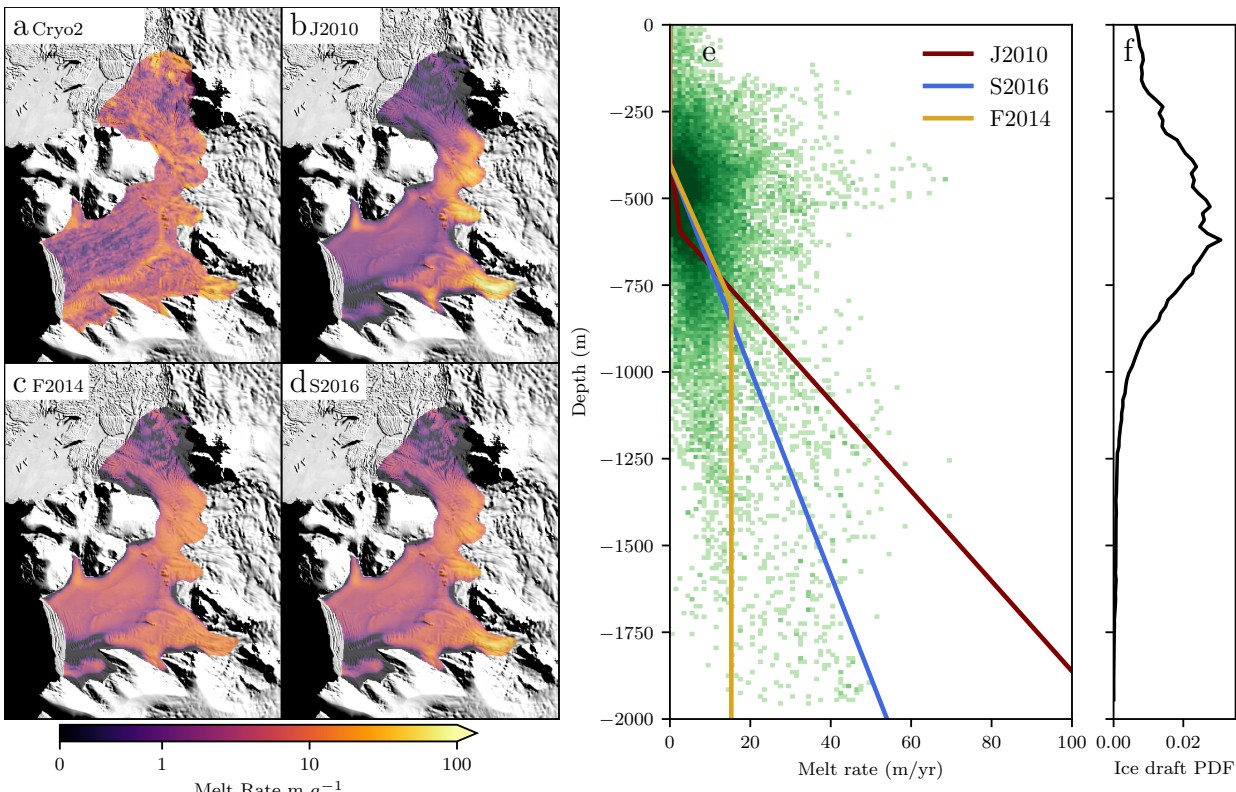

**Figure 2. Melt forcings used for modeling. a-d.** Distribution of melt rates at the beginning of simulations using Cryosat-
inferred rates from Gourmelen et al. (2017), and parameterizations from Joughin et al. (2010), Favier et al. (2014), and Shean
(2016) respectively. **e.** Scaled parameterizations (colored lines) plotted over green points showing Cryosat-derived distribution of
melt. Darker colors indicate greater area with a given combination of depth and melt rate. **f.** PDF of depths; this indicates the total
area at each depth, showing how shelf-total melt rates are most sensitive to melt rates between ~250 and 800 meters.

| | Model physics | Melt dist. | Melt intensity | Enhancement in margins | Forced ungrounding | Sliding Law | Sim. len. (years) | >15 km retreat |
|---|---|---|---|---|---|---|---|---|
| 1 | FS | J2010 | 1Obs | 1 | No | Schoof | 23 | Yes |
| 2 | SSA | J2010 | 1Obs | 1 | No | Schoof | 104 | Yes |
| 3 | SSA | J2010 | 2Obs | 1 | No | Schoof | 25 | Yes |
| 4 | SSA | J2010 | 1Obs | 4 | No | Schoof | 104 | Yes |
| 5 | SSA | J2010 | 1Obs | 1.8 | No | Schoof | 50 | Yes |
| 6 | SSA | J2010 | Control | 4 | No | Schoof | 50 | No |
| 7 | SSA | J2010 | 1Obs | 1 | 5 years | Schoof | 50 | Yes[†] |
| 8 | SSA | J2010 | 1Obs | 1 | 18 years | Schoof | 50 | Yes*[†] |
| 9 | SSA | J2010 | 1Obs | 1 | No | Weertman | 50 | Yes |
| 10 | SSA | F2014 | 1Obs | 1 | No | Schoof | 104 | No |
| 11 | SSA | F2014 | 2Obs | 1 | No | Schoof | 25 | Yes |
| 12 | SSA | F2014 | 1Obs | 4 | No | Schoof | 50 | No |
| 13 | SSA | F2014 | Control | 4 | No | Schoof | 50 | No |
| 14 | SSA | F2014 | 1Obs | 1 | 5 years | Schoof | 50 | No[†] |
| 15 | SSA | F2014 | 1Obs | 1 | 18 years | Schoof | 50 | Yes* |
| 16 | SSA | F2014 | 1Obs | 1 | No | Weertman | 50 | No |
| 17 | SSA | S2016 | 1Obs | 1 | No | Schoof | 104 | No |
| 18 | SSA | S2016 | 2Obs | 1 | No | Schoof | 25 | Yes |
| 19 | SSA | S2016 | 1Obs | 4 | No | Schoof | 104 | Yes |
| 20 | SSA | S2016 | Control | 4 | No | Schoof | 50 | No |
| 21 | SSA | S2016 | 1Obs | 1 | 5 years | Schoof | 50 | Yes[†] |
| 22 | SSA | S2016 | 1Obs | 1 | 18 years | Schoof | 50 | Yes* |
| 23 | SSA | S2016 | 1Obs | 1 | No | Weertman | 50 | No |
| 24 | SSA | Cryo2 | 1Obs | 1 | No | Schoof | 104 | No |
| 25 | SSA | Cryo2 | 2Obs | 1 | No | Schoof | 25 | No |
| 26 | SSA | Cryo2 | 1Obs | 4 | No | Schoof | 50 | No |
| 27 | SSA | Cryo2 | Control | 4 | No | Schoof | 50 | No |
| 28 | SSA | Cryo2 | 1Obs | 1 | 5 years | Schoof | 50 | No[†] |
| 29 | SSA | Cryo2 | 1Obs | 1 | 18 years | Schoof | 50 | Yes* |
| 30 | SSA | Cryo2 | 1Obs | 1 | No | Weertman | 50 | No |

**Table 1. Summary of model inputs.** Model physics and inputs are summarized in the first six columns**.** The last column
indicates whether the Smith Glacier grounding line retreated over 15 km within the simulation, with starred entries indicating that
retreat was explicitly forced. Daggers indicate that some grounding-line retreat continued beyond the period of explicit forcing.

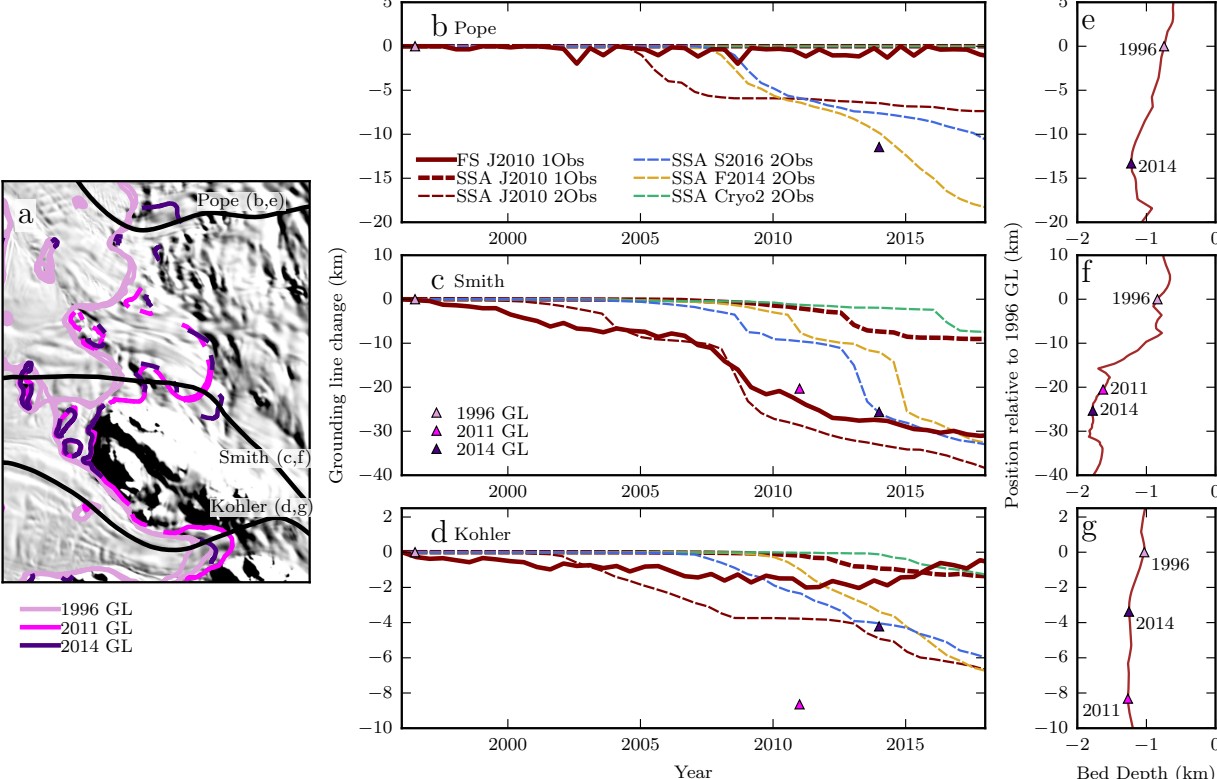

**Figure 3. Sensitivity of change in grounding line position to melt distribution and intensity. a.** Flowlines used for evaluation of grounding line retreat (black). Pink and purple lines indicate observed grounding line positions (Rignot et al., 2014; Scheuchl et al., 2016). **b-d.** Modeled and observed grounding line position along the centerlines of Pope, Smith, and Kohler Glaciers respectively for different model simulations. Zero indicates no change since 1996, negative values indicate retreat. Line colors indicate melt distribution: J2010 (maroon), S2016 (blue), F2014 (gold), and Cryo2 (green). Line thickness indicates melt intensity: thick for 1Obs, thin for 2Obs. Line style indicates full-Stokes (solid) or shallow-shelf model (dashed). Simulations that display less than 2 km of grounding-line retreat on all centerlines are not shown. Triangles indicate observations of grounding line position, with colors corresponding to lines in a. **e-g.** Bed elevations vs distance from 1996 grounding line along the centerlines of Pope, Smith, and Kohler Glaciers respectively. Vertical scale matches panels b-d. Purple triangles again indicate observed grounding line positions through time.


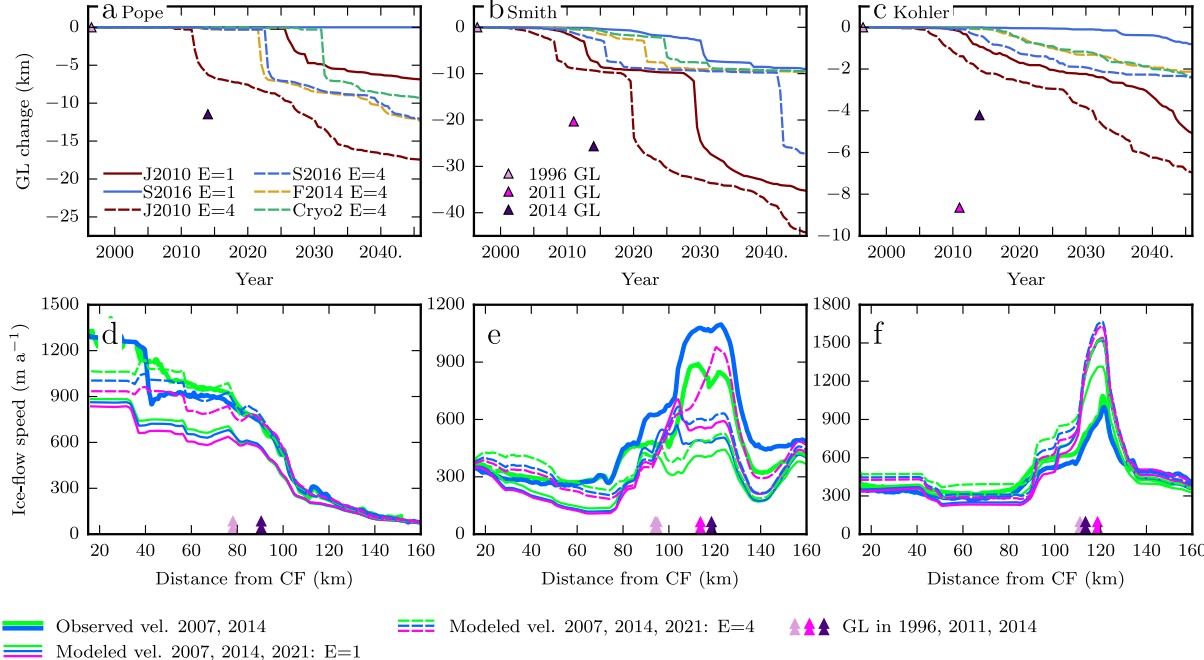


**Figure 4. Effect of marginal weakening on grounding-line position and velocity. a-c.** Modeled grounding-line position through
time along Pope, Smith East, and Kohler along flowlines shown in Figure 1. All simulations used 1Obs melt intensity. Colors
indicate the melt forcing as in Figure 3. Solid line indicates no weakening, and dashed line indicates 4x enhancement within 10 km
of the ice-shelf margins. Triangles show observed grounding line position (Rignot et al., 2014; Scheuchl et al., 2016). **d-f.** Velocity
along flowlines corresponding to upper panels, with all simulations now using the J2010 melt parameterization. Color of line
indicates the year (blue for 2007, green for 2014, pink for 2021). Thick lines show observations. Thinner lines show model results
(using the J2010 melt parameterization), with dashed and solid patterns corresponding to the upper panels. Arrows at bottom
indicate observed grounding-line position through time.

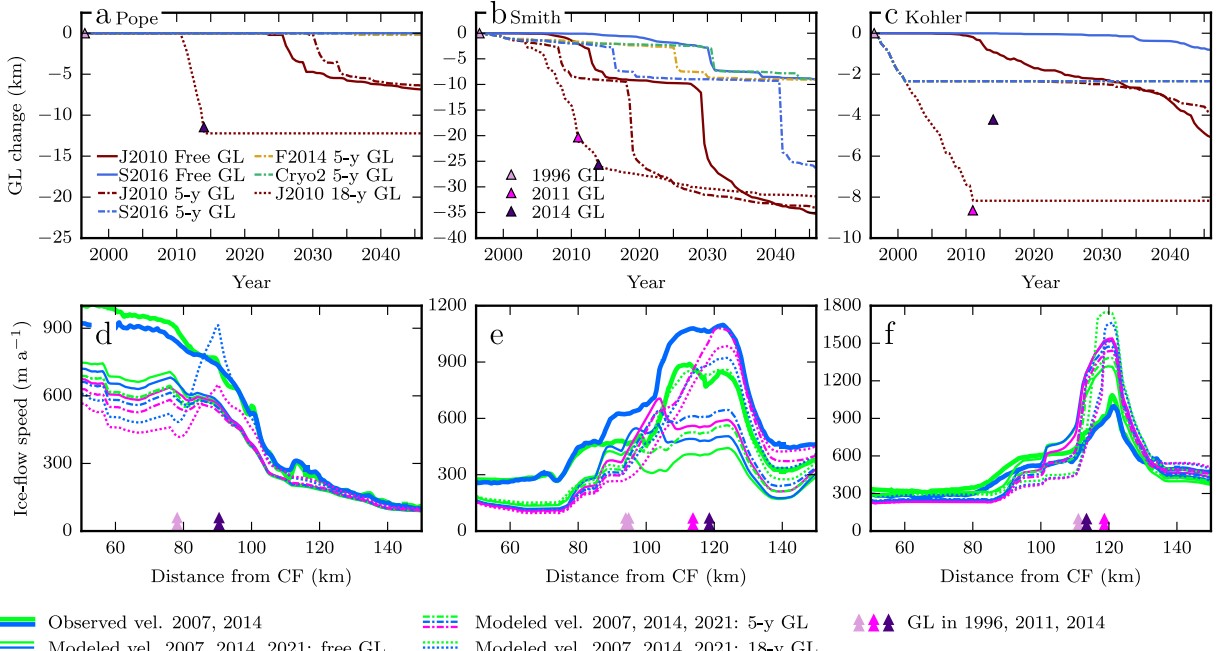


**Figure 5. Grounding line and speed changes resulting from forced ungrounding. a-c.** Modeled grounding-line positions along centerlines of Pope, Smith East, and Kohler centerlines, respectively, from Figure 1. All simulations used 1Obs melt intensity with no marginal enhancement. Line style indicates how the grounding line was treated: solid line for freely evolving grounding line, dashed line for forced ungrounding for 18 years (1996-2014), and dash-dot for forced ungrounding for 5 years only (1996-2001). Triangles indicate observed grounding-line positions through time (Rignot et al., 2014; Scheuchl et al., 2016).. Simulations with no change in grounding-line position after the forced ungrounding are not shown. **d-e.** Observed and modeled ice speed along centerlines from upper panels, with line color indicating year as in Figure 4. Thick lines show observations. Thinner lines show model simulations (using J2010 melt distribution) with line style indicating ungrounding scheme as in a-c. Triangles at bottom indicate the observed grounding-line position in different years; the effect of forced ungrounding on modeled ice speed is generally restricted to the area around the grounding line where the surface remains relatively steep while basal resistance is removed.

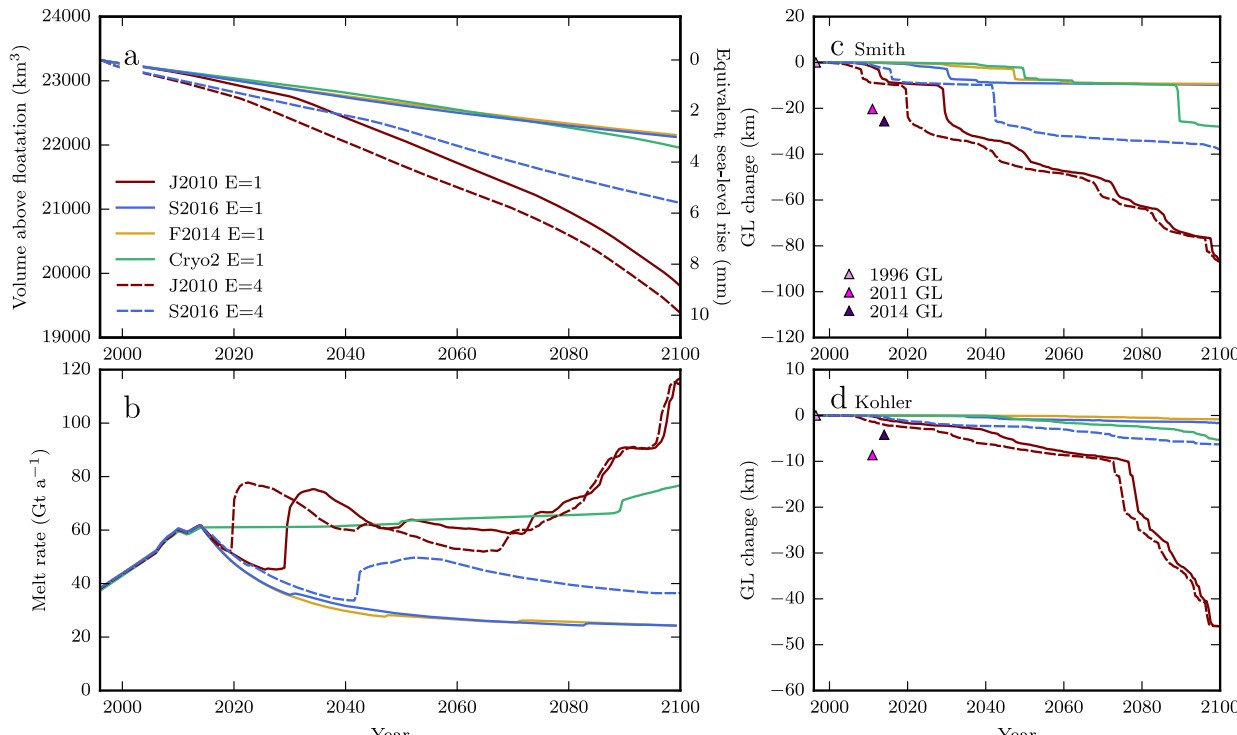

830

**Figure 6. Results of centennial-scale model simulations. a.** Volume above floatation in the Smith, Pope, Kohler catchment and equivalent sea-level rise through time for extended simulations. All runs use 1Obs melt intensity. Color of line indicates melt distribution as in previous figures. Solid line corresponds to shallow-shelf model, and dashed line shows shallow-shelf model with enhanced margins. The differences in volume during the period including forcing result from different ice-flow speeds causing different calving rates. **b.** Melt rate through time. Runs are forced to observations through 2014, so melt rates correspond through this period, then diverge since the scaling of the melt parameterization is fixed at the 2014 value. Note that melt rates do not directly cause loss of volume above floatation since some melt distributions cause melt of the shelves without substantial loss of grounded ice. **c-d.** Grounding-line position change through time along Smith and Kohler centerlines, respectively, from Figure 1. Purple triangles again show observed grounding-line positions through time (Rignot et al., 2014; Scheuchl et al., 2016).

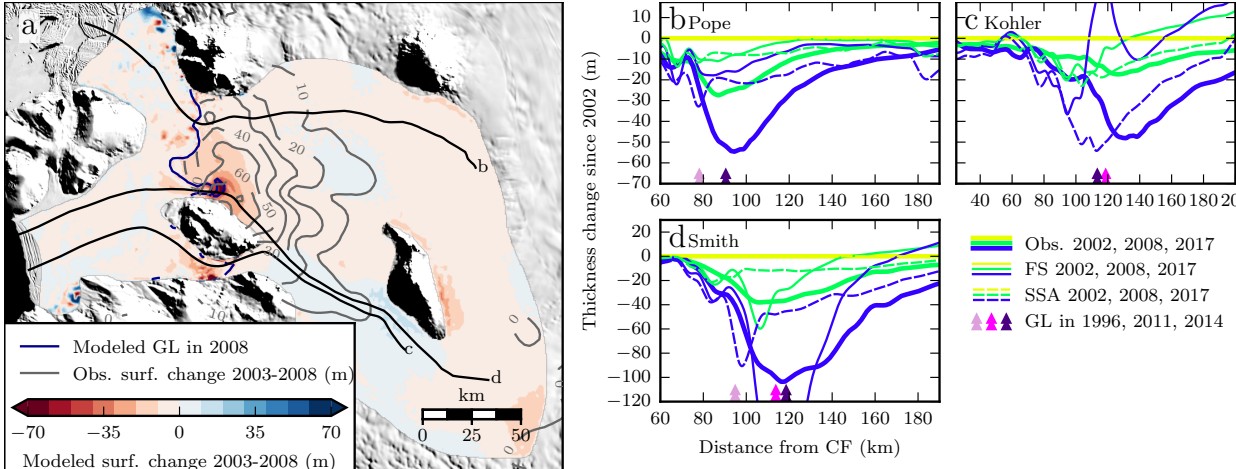


**Figure 7. Modelled and observed thinning during the ICESat era (2003-2008). a.** Spatial distribution of thinning using the
shallow-shelf model with J2010 1Obs melt. Colors indicate modelled thickness change while grey contours indicate observations.
Black lines show flowlines as in other figures. Thin, blue line shows the modelled grounding line in 2008. **b-e.** Thinning through
time along flowlines. Color indicates the year. Thin lines show model, thick lines show data derived from Operation IceBridge
altimetry, ICESat-1, and WorldView/GeoEye DEMs. Triangles indicate grounding-line position (Rignot et al., 2014; Scheuchl et
al., 2016).