# Peer review of "Melt at grounding line controls observed and future retreat of Smith, Pope, and Kohler Glaciers"

_The Cryosphere, 2019_

## Referee Comment (RC1) · Martin Truffer (Referee) · 5 Jun 2019

This paper presents models of recent thinning and grounding line retreat of the glaciers feeding the Dotson and Crosson ice shelves. The models are also used to address possible future scenarios for this system of glaciers. The paper is hugely relevant. While a lot of attention is currently focused on the neighboring Thwaites Glacier it can easily be forgotten that the Pope/Smith/Kohler glaciers have undergone some of the largest changes observed anywhere on the planet. In addition, it has the potential to affect ice evolution in the larger area, as thinning can spread inland rapidly and lead to divide migration, with consequences for this entire sector of the ice sheet.

[Figure]

I do recommend publication in TC for this paper, but I also have a few general comments that I hope will be useful.

1) Generally, the paper could do a better job in outlining what works well in these models and what doesn't. This could be accomplished by a slight reorganisation of the Discussion and some expansion of the Conclusions. Otherwise, it is easy to read this paper and get sidetracked by model-data mismatches. This starts with Figure 1: Fig. 1b-d show velocity model-data mismatches that are quite large. It is easy for a reader to then be skeptical of any conclusion reached in the paper. I suggest that the paper first emphasises the conclusions that are most solidly supported and then discusses all the qualifications. For example, continued mass loss over the next century of order >6 mm sea level seem inescapable. Grounding line position is fiendishly difficult to get right and varies a lot between models. Etc.

2) What is the critererion for the choice of models for the prognostic simulation? It seems like you don't hold much faith in some of these. Could you more clearly outline, which range is most realistic, given the model performance over the period of observations.

3) I would love to see a bit more discussion on initialization. You generally do a good job outlining the challenges. Is there a way to assess how important the initial temperature distribution is? You make a steady state calculation here; if I read it correctly. For example, when you invert for flow rate factors over the ice shelf, how does that compare to the derived temperature distribution? Also, one measure of success for initialization is to look at thinning rates. How well do the models do with that?

4) How is calving at the ice shelf front handled?

5) Could you comment a bit more on the choice of multiplying observed melt rates rather than multiplying parameters as in prior studies (line 224-230). What are the benefits of this choice?

6) Parts of the discussion on weakening (l.383-394) reads a bit odd in the sense that it sounds like you discuss weakening of margins as a possible cause. But what would cause weakening in the first place? Ice doesn't just get 5 deg warmer or more anisotropic; there would have to be some other driver. So weaker margins could lead to an amplification of an otherwise triggered change. Some rewording would clarify that.

7) Figures are a bit hard to read with small fonts, at least on a printed out copy. I would prefer a Figure 1 that is more of an overview. In particular, having results in there already (velocities) is actually distracting

8) This is a bit of a repeat of comment 1): What should the reader take away from figures such as Fig. 7? I can look at it and say that this model is terrible: on Pope the largest thinning is off by a factor of 2 in the best case. Similar things could be said for grounding line positions and velocities. But that is obviously not your main point. Help the reader a bit in what you consider the successes and challenges of this modeling effort. I think a bit of a restructure of Discussions would go a long ways here.

Small edits:

l.30: I would say 'peaked temporarily'. There is no reason that this would have to remain a one-time occurrence.

l.175: I think this is not quite correct. The effective pressure assumption here essentially implies infinite hydraulic conductivity (a flat water table). The implication is that any sort of pressure gradient that is required to drain subglacial water means that water pressure further away from the grounding line needs to be higher, which extends Coulomb like deformation inland. Therefore the model is likely to underestimate inland velocity response.

l.203: ... comparison BETWEEN modeled ...

l.277: ... compared TO the observed ...

l.324/25: Why those particular choices (see also comment 2) above)

l.445: to -> from

l.449: Where is the Haynes Glacier (maybe show in Fig. 1?)

l.463: There are A variety ...

l.509: error -> errors

l.516-519: Also, hydraulic gradients would lead to lower effective pressure inland, as per comment above.

l.526: .. as much AS the ...

l.526: How do you know that?

l.530: its -> it

l.531: ... we did NOT have ...

l.567: 'relatively modest' is in the eye of the beholder, you're describing some major changes here with global impacts from a single basin

Conclusions could be expanded a bit.

l.759: one of the 'thin for 2Obs' should be 'thick for 1Obs'

Martin Truffer
* * *

---

## Referee Comment (RC2) · Anonymous Referee #2 · 15 Jun 2019

General Comments

In this paper, the authors present the results of from a model of the Smith, Pope, and Kohler glaciers using varying sub-shelf melt forcings and marginal shelf weakening. The experimental design is thoughtfully considered and thorough, with a large number of combinations of model setups tested. Any limitations to the model are also thoroughly discussed at the end of the paper. After backtesting the model over the previous 25 years and comparing to observations, the model is run forward in time to predict the retreat of the grounding lines of Smith, Pope, and Kohler glaciers over the next century. The authors find that the glaciers are likely to contribute at least 6mm of sea level rise over the next 100 years. They also predict that Smith glacier could retreat to the ice divide with the Thwaites glacier catchment within the next 100 years, further

undermining the ice sheet in the Thwaites drainage. This is an interesting and important result that will likely be of special interest to the broader community of researchers studying the Amundsen Sea region of the Antarctic Ice Sheet.

I recommend that the paper be accepted for publication subject to the following comments/corrections.

Specific Comments

P. 2, lines 40-41: If the grounding line is retreating to deeper seabed, the warm water will need to flow down over the shallower seabed to get to it, meaning that the grounding line is no more vulnerable to warm ocean water than it was before. Do you mean that the grounding line is more vulnerable to melting due to the reduction in the freezing point with depth?

P. 7, line 212: It would be interesting to know how the Cryo2 melt rates compare to flux divergence melt rates for the 2010-2016 period. Have you looked into this?

P. 8, line 277: "5-km retreat" looks more like ∼1km retreat to me.

P. 9, line 308: Referral to Figure 4a-c, but control melt results aren't shown.

P. 9, line 314: Referral to Figure 5a-c to see stabilization after 5 year forced ungrounding, but the first five years of grounding line retreat are covered by the figure label (especially for Smith).

P. 9, lines 314-315: "retreat subsequently ensues on each of the three glaciers..." This doesn't appear to be true for S2016 melt on Pope or Kohler.

P. 10, lines 320-322: "...by the end of the 50-year simulations..." To my eye, the only one that consistently approximates the J2010 free GL results is J2010 5-yr GL.

P. 10, line 335: Define "significant" (>40km?)

P. 11, line 357: "...those with the J2010 melt parameterization..." (with the exception of

the control melt-scaling)

P. 11, lines 375-378: "While our rescaling..." What does this mean for Cryo2 melt rates from 1996-2010? The Cryo2 melt distribution also doesn't allow the ice shelf to deepen into melt as you go back in time – does this mean that melt rates near the grounding line would be comparatively high for the Cryo2 distribution in 1996?

P. 12-13, Section 4.1.2: Should at least part of this be in the Results section?

P. 13, lines 423-425: It looks like the SSA simulations thin too little for all three glaciers upstream of their grounding lines.

Figure 2: You say "shelf total melt rates are most sensitive to melt rates between $\sim$250 and 600 meters." Why not choose two depths where the PDF values are equal, e.g. $\sim$250 and $\sim$800m?

Table 1: Simulation number 8 has an asterisk indicating that the retreat was entirely forced, but in Figure 5 we can see that the retreat continues beyond the 18y grounding line for Smith glacier.

Figure 4: The important first 5 years of grounding line retreat are covered by the plot labels for Smith and Kohler glaciers. Similar for Figure 5.

Technical Corrections

P. 2, lines 49-50: I found this wording confusing – it sounds like "committed" is a verb. Maybe reword it?

P. 4, line 121: "allows us to"

P. 5, line 155: "We ran a suite"

P. 6, line 178: I think this should say something like "...errors due to the assumption are alleviated through choice of the sliding coefficient..."

P. 7, line 232: Perhaps a matter of taste, but I like the word "margins" more than "edge."

Edge makes me think of the calving front.

P. 8, line 279: 20011 → 2011

P. 10, line 321: should say "...using all four melt distributions...?"

P. 10, line 329: I think the referral to Figure 6 is repetitive, given that the whole paragraph is referring to the figure. You could make it a referral to Figures 6c-d if you want to be specific.

P. 11, line 374: mix → mixes

P. 11, line 379: instantiate → induce (or trigger)?

P. 12, line 391: "simulations with marginal"

P. 12, line 401: indicate → indicates

P. 12, line 405: 19 → 21

P. 14, line 458: across → along?

P. 14, line 463: "There is a variety..."

P. 14, line 470: complicate → complicates

P. 14, lines 482-483: The way this is worded makes it sound like it was the SSA simulations that required HPC. Maybe say "... allowing the use of local workstations rather than requiring high-performance computing resources."

P. 15, line 509: have → "has" or "may have"

P. 15, line 528: "not be an indication" ?

P. 16, line 531: "we did not have"

Figure 1: I assume there should be a box showing the study area on the map of Antarctica – I can't see one on my printed copy.

[Figure]

Figure 3, line 759: "thick for 1Obs, thin for 2Obs."

Figure 5, line 779: double periods

Figure 5, line 782-782: "Color of the line indicates year" already stated in line 781.

Figure 6: Axis labels are overlapping for 6a and 6c.

Figure 6, line 790: "difference...result" → "difference ... results" or "differences ... result"

---

## Author Comment (AC1) · 1 Aug 2019

This paper presents models of recent thinning and grounding line retreat of the glaciers feeding the Dotson and Crosson ice shelves. The models are also used to address possible future scenarios for this system of glaciers. The paper is hugely relevant. While a lot of attention is currently focused on the neighboring Thwaites Glacier it can easily be forgotten that the Pope/Smith/Kohler glaciers have undergone some of the largest changes observed anywhere on the planet. In addition, it has the potential to affect ice evolution in the larger area, as thinning can spread inland rapidly and lead to divide migration, with consequences for this entire sector of the ice sheet.

I do recommend publication in TC for this paper, but I also have a few general comments that I hope will be useful.
*Thank you for the careful read and helpful comments; we have incorporated them as described below.*

1) Generally, the paper could do a better job in outlining what works well in these models and what doesn't. This could be accomplished by a slight reorganisation of the Discussion and some expansion of the Conclusions. Otherwise, it is easy to read this paper and get sidetracked by model-data mismatches. This starts with Figure 1: Fig. 1b-d show velocity model-data mismatches that are quite large. It is easy for a reader to then be skeptical of any conclusion reached in the paper. I suggest that the paper first emphasises the conclusions that are most solidly supported and then discusses all the qualifications. For example, continued mass loss over the next century of order >6 mm sea level seem inescapable. Grounding line position is fiendishly difficult to get right and varies a lot between models. Etc.
*It is certainly a challenge to distill the most valuable aspects of the modeling. Following this comment, we restructured the text at several places so as to try to emphasize the most strongly supported conclusions while still presenting the necessary caveats, which involved four main changes:*

1. *We have switched figure 1 to be data only and have removed the panels of modeled speed since the changes in velocity can also be gathered from later figures.*

2. *For the discussion, we have tried to emphasize the comparison to observed grounding-line positions, because this comparison is the main tool by which we evaluate the effects of different model parameters upon the results. We moved the discussion century-scale simulations to immediately follow this section, since the grounding-line comparison provides the necessary framework for understanding which of those longer-term simulations we consider most likely. (see sections 4.1 and 4.2 of the revised manuscript)*

3. *To begin the comparison with ice velocities and thinning, we explicitly acknowledge the substantial mismatch, but emphasize that the mismatch nevertheless allows us to learn about what important processes the model might be missing and in what direction it may err.*

4. *We have expanded the conclusions section to reflect these most inescapable aspects of the model simulations.*

2) What is the criterion for the choice of models for the prognostic simulation? It seems like you don't hold much faith in some of these. Could you more clearly outline, which range is most realistic, given the model performance over the period of observations. *We chose which models to run with three criteria: 1. All the runs were 1Obs so as to have the most realistic melt rates. 2. The simulations with substantial retreat were all represented and 3. We spanned even some simulations without much retreat, in case retreat eventually continued beyond the time span of the short simulations. We have added a brief mention of this in the methods* "were chosen to represent a range of retreat rates, some realistic and some slower than observed, and all used realistic melt rates." *and a reminder in the results* "chosen to represent a range of retreat scenarios with realistic melt intensity."

*We have also changed the discussion to start by explicitly reminding the reader that we only consider 3/6 of these simulations to be reasonable matches to observations. The new pieces of text are:*
"The centennial-scale simulations can be broadly categorized that emulate observed grounding-line retreat (i.e. display more than 35 km of retreat) and those that retreat less than observations. Those simulations that emulate retreat (2, 4, and 19 in Error! Reference source not found.) all continue to produce retreat into the future."
*and*
"Thus, these three simulations suggest that these glaciers will likely contribute 6 mm of sea-level rise over the coming century, even if shelf-integrated melt rates remain at about their levels in recent years."

3) I would love to see a bit more discussion on initialization. You generally do a good job outlining the challenges. Is there a way to assess how important the initial temperature distribution is? You make a steady state calculation here; if I read it correctly. For example, when you invert for flow rate factors over the ice shelf, how does that compare to the derived temperature distribution? Also, one measure of success for initialization is to look at thinning rates. How well do the models do with that?

*Yes, it is indeed a steady state calculation. We frame the inversion in terms of the enhancement factor, so the most straightforward measure of how close this is to the temperature distribution is to look at where these values are close to one. The initial inversion results over the shelves (prior to relaxation) were previously published (Lilien*

*et al., 2018, figure 5); generally, the enhancement factor is ~1.5 or less, with the temperature being largely sufficient to capture the 1996 velocities (though greater enhancement was needed to accurately model the velocities in the 2010s). However, this does not necessarily imply that the modeled temperature is correct, and it may just be a weak sensitivity of the modeled velocities to the viscosity. Regardless, it appears that temperature alone does reasonably well on the shelves. We have added* "this initial set of inversions is also described in Lilien et. al, 2018, where plots of the inferred enhancement are shown".

*It is a much tougher question to assess whether the temperature distribution of grounded ice is sufficiently uncertain to adversely affect our results, as we may have compensating errors between the temperature distribution and basal slipperiness field. While some previous work has simultaneously inferred ice viscosity and basal slipperiness, we choose not to do so due to the potential for non-unique inference of the fields. While thinning rates may theoretically indicate something about model initialization accuracy, in practice they can simply indicate something about the accuracy of ice-thickness measurements. For example, even if a model had "true" basal friction and effective viscosity but had a 40-m error in bed elevation, we would expect substantial thinning/thickening for mass conservation. We have added a paragraph describing this assessment:* "While it is difficult to assess whether the model accurately represents the true temperature, enhancement, and basal slipperiness fields, modeled thinning rates at the end of relaxation give an indication of model self-consistency. Conversely, the total change in surface height during relaxation gives a misfit between the model and available data (though in part that relaxation may be compensating for errors in the data). Here, relaxation resulted in local changes of up to 100 m near Kohler Glacier's grounding line and changes of at most 50 m elsewhere. While most of the change during relaxation can potentially be attributed to errors in ice thickness caused by uncertainty in the bed elevation, the large change on Kohler likely indicates that the surface elevations were also incorrect in that area. Because determining the surface elevations at initialization required some extrapolation using longer term thinning rates (see Lilien et al., 2018 for details), this misfit is not surprising and may reflect a change in the spatial pattern of thinning during 1996-2003. At the end of the relaxation, thickness change rates were reduced to <10 m a$^{-1}$, which is smaller than the observed rate of thickness change, except on Kohler Glacier where ~30 m a$^{-1}$ of thickening persisted. While this is still a large rate of elevation change on Kohler, we were forced to choose between accepting Kohler's unrealistic imbalance and possibly relaxing away the real imbalance on Smith and Pope. The potential effects of the resultant transients upon the modeled retreat of Kohler are revisited in Section Error! Reference source not found.."

**4) How is calving at the ice shelf front handled?**
*Because the calving fronts have remained conveniently near the ends of their embayments, we do not explicitly model calving but simply use a sea-pressure boundary condition at the end of the embayment. We recognize that this of interest for readers, and have added:* "Calving was not explicitly modeled, but instead ocean pressure was applied on the downstream boundary at the mouth of the ice shelves' embayments where ice is

allowed to flow out. This boundary condition would remain accurate for an advance since ice tongues extending beyond embayment walls do not provide additional back stress, but, if substantial ice loss caused the calving front to retreat behind the embayment walls, it could potentially result in underestimating ice loss during retreat."

5) Could you comment a bit more on the choice of multiplying observed melt rates rather than multiplying parameters as in prior studies (line 224-230). What are the benefits of this choice?
*The main benefit is preventing unrealistically large losses during a ramp-in period, while the primary drawback is the requirement to have the data available to constrain these rates. We do not want to elaborate too much here since there is a full paragraph in the discussion, but have added the brief description "*but this choice was mainly made to limit melt rates to realistic values during the period with observations"

6) Parts of the discussion on weakening (l.383-394) reads a bit odd in the sense that it sounds like you discuss weakening of margins as a possible cause. But what would cause weakening in the first place? Ice doesn't just get 5 deg warmer or more anisotropic; there would have to be some other driver. So weaker margins could lead to an amplification of an otherwise triggered change. Some rewording would clarify that.
*Indeed there needs to be a trigger, and the wording was ambiguous in this section. We have rephrased at several points, and it now reads: "*While snapshot inversions for ice-shelf viscosity in 1996, 2011, and 2014 indicate some weakening of Crosson Ice Shelf (Lilien et al., 2018), this weakening cannot be definitively identified as having been caused by a particular process (e.g. loss of a pinning point or rifting). Thus, we are unable to identify if the weakening of the margins was triggered by grounding-line retreat itself or was externally triggered and helped initiate grounding-line retreat. We consider it unlikely, regardless of their trigger, that"

7) Figures are a bit hard to read with small fonts, at least on a printed out copy. I would prefer a Figure 1 that is more of an overview. In particular, having results in there already (velocities) is actually distracting
*We agree that these are hard to read in the version online. Font sizes were all in line with the publisher recommendations when the figures were produced, but shrunk substantially due to different margin widths between the template files and what is expected for final, published versions, and also due to shrinking by the publisher after submission in order to fit logos at the top of each page for the discussion version. We apologize for the difficulty in reading them, but have left the font sizes as-is for resubmission since we expect the problem to be remedied by using the full-size figures at production.*

*We have changed figure 1 to be data alone.*

8) This is a bit of a repeat of comment 1): What should the reader take away from figures such as Fig. 7? I can look at it and say that this model is terrible: on Pope the largest thinning is off by a factor of 2 in the best case. Similar things could be said for

grounding line positions and velocities. But that is obviously not your main point. Help the reader a bit in what you consider the successes and challenges of this modeling effort. I think a bit of a restructure of Discussions would go a long ways here.
*Addressed along with comment 1.*

Small edits:

l.30: I would say 'peaked temporarily'. There is no reason that this would have to remain a one-time occurrence.
*Done*

l.175: I think this is not quite correct. The effective pressure assumption here essentially implies infinite hydraulic conductivity (a flat water table). The implication is that any sort of pressure gradient that is required to drain subglacial water means that water pressure further away from the grounding line needs to be higher, which extends Coulomb like deformation inland. Therefore the model is likely to underestimate inland velocity response.
*Yes, this is probably a better description of the assumption, and one of multiple reasons that this parameterization may underestimate Coulomb-like behavior. We have changed the text to:*
"This assumption is valid for infinite hydraulic conductivity, while in reality inland water pressures may be higher due to finite hydraulic conductivity, which would lead to this parameterization underestimating the extent of Coulomb-like behavior"

l.203: ... comparison BETWEEN modeled ...
*Fixed*

l.277: ... compared TO the observed ...
*Fixed*

l.324/25: Why those particular choices (see also comment 2) above)
*We elaborate more elsewhere as described above, but here we have added* "chosen to represent a range of retreat scenarios with realistic melt intensity"

l.445: to -> from
*Fixed*

l.449: Where is the Haynes Glacier (maybe show in Fig. 1?)
*Since it is mostly clipped from figure 1, we have instead specified in the text that we refer to loss of buttressing at the outer right corner of Crosson.*

l.463: There are A variety ...
*Fixed*

l.509: error -> errors
*Fixed*

l.516-519: Also, hydraulic gradients would lead to lower effective pressure inland, as per comment above.
*Changed to* "ignoring hydraulic gradients and limiting Coulomb (plastic) behavior to near the grounding line"

l.526: .. as much AS the ...
*Fixed*

l.526: How do you know that?
*We have made this more tentative:* "it is possible bed errors alone could change the timing of retreat by as much as the model-data mismatch."

l.530: its -> it
*Done*

l.531: ... we did NOT have ...
*Fixed*

l.567: 'relatively modest' is in the eye of the beholder, you're describing some major changes here with global impacts from a single basin
*Changed to:* "modest compared to some other Antarctic catchments"

Conclusions could be expanded a bit.
*Done. Changes are described in response to General Comment 1.*

l.759: one of the 'thin for 2Obs' should be 'thick for 1Obs'\
*Thanks*

**Response to anonymous reviewer #2**

General Comments

In this paper, the authors present the results of from a model of the Smith, Pope, and Kohler glaciers using varying sub-shelf melt forcings and marginal shelf weakening. The experimental design is thoughtfully considered and thorough, with a large number of combinations of model setups tested. Any limitations to the model are also thoroughly discussed at the end of the paper. After backtesting the model over the previous 25 years and comparing to observations, the model is run forward in time to predict the retreat of the grounding lines of Smith, Pope, and Kohler glaciers over the next century. The authors find that the glaciers are likely to contribute at least 6mm of sea level rise

over the next 100 years. They also predict that Smith glacier could retreat to the ice divide with the Thwaites glacier catchment within the next 100 years, further undermining the ice sheet in the Thwaites drainage. This is an interesting and important result that will likely be of special interest to the broader community of researchers studying the Amundsen Sea region of the Antarctic Ice Sheet.

I recommend that the paper be accepted for publication subject to the following comments/corrections.

*We thank the reviewer for the careful read and comments. Specific points are addressed below.*

Specific Comments

P. 2, lines 40-41: If the grounding line is retreating to deeper seabed, the warm water will need to flow down over the shallower seabed to get to it, meaning that the grounding line is no more vulnerable to warm ocean water than it was before. Do you mean that the grounding line is more vulnerable to melting due to the reduction in the freezing point with depth?

*There are several reasons besides freezing-point depression that deepening a grounding line can increase melt rates. First, at the shallowest point in the bathymetry, shallowing draft thickens the water column and permits greater access of water to contact the ice. Recent modeling shows that the ice-bottom topography plays an important role in how water accesses the cavity (Goldberg et al., 2019). Moreover, deeper grounding lines generally imply greater sub-shelf area below any given depth, again creating more potential melt independent of the local melting point (this is stated directly in Jenkins et al., 2018, which we cite at this point in the text). We believe the statement is well supported, and have left it as-is.*

P. 7, line 212: It would be interesting to know how the Cryo2 melt rates compare to flux divergence melt rates for the 2010-2016 period. Have you looked into this?

*We have looked into this on a shelf-averaged scale for Dotson, where they agree within error (7.7±1.3m yr⁻¹ vs. 6.1±0.7myr⁻¹). We have added mention in the text that agree to within errors.*

P. 8, line 277: "5-km retreat" looks more like ~1km retreat to me.

*We are guessing that this is a typo, and the reviewer meant 10-km retreat. Regardless, it is a good catch. Depending on the exact flowline used, it ranges from ~5-12 km; since the flowline used in the figures has it near 10 km, we have switched it to 10 km here.*

P. 9, line 308: Referral to Figure 4a-c, but control melt results aren't shown.

*This referred to an earlier version of the figure where we had included more different simulations that were subsequently removed for readability. We have deleted the reference.*

P. 9, line 314: Referral to Figure 5a-c to see stabilization after 5 year forced ungrounding, but the first five years of grounding line retreat are covered by the figure label (especially for Smith).
*Yes, the label was poorly placed. We have moved the figure label to make this visible.*

P. 9, lines 314-315: "retreat subsequently ensues on each of the three glaciers..." This doesn't appear to be true for S2016 melt on Pope or Kohler.
*Good point, the text was in error. We have fixed this to only reference Smith glacier.*

P. 10, lines 320-322: "...by the end of the 50-year simulations..." To my eye, the only one that consistently approximates the J2010 free GL results is J2010 5-yr GL.
*Our language was imprecise. We have made this a simple, quantitative statement to avoid the ambiguity. It is now* "leaving the grounding lines within 5 km of their 2014 positions"

P. 10, line 335: Define "significant" (>40km?)
*We added >50 km as a parenthetical. At several points in the text, we switched the language from "significant" to "substantial" to avoid implications of statistical significance.*

P. 11, line 357: "...those with the J2010 melt parameterization..." (with the exception of the control melt-scaling)
*Another good catch. We have rephrased to* "J2010 melt parameterization with 1Obs or 2Obs melt, irrespective of marginal weakening"

P. 11, lines 375-378: "While our rescaling..." What does this mean for Cryo2 melt rates from 1996-2010? The Cryo2 melt distribution also doesn't allow the ice shelf to deepen into melt as you go back in time – does this mean that melt rates near the grounding line would be comparatively high for the Cryo2 distribution in 1996?
*The rates used to force the model are low there because the areas of high melt in 2010-2016 are generally in newly ungrounded area. We have changed the text to clarify how this mismatch in timing and forcing are affect the results. It now reads:*
"The stable grounding-line position found by forcing the model with Cryo2 melt (Gourmelen et al., 2017) may result from underestimation of melt near the grounding line in 1996, either due to the difficulty of using satellite altimetry to infer melt rates in an area not in hydrostatic equilibrium (Fricker and Padman, 2006; Rignot, 1998) or due to a change in distribution of melt between 1996 and 2010. Since melt rates were inferred over 2010-2016, if melt were highest near the grounding line in 1996 but subsequently the area of peak melt moved upstream, the 2010-2016 rates may be much lower those in 1996 near the grounding line at that time. This mismatch in observation time and model forcing could have then resulted in the model never beginning to retreat into areas of concentrated melt. Moreover, even once retreat was triggered, the inferred melt rate beneath areas that ungrounded during 2010-2016 mixes periods of no melt and more intense melt, thus causing underestimation of the annual-average melt during the periods when the ice was ungrounded."

P. 12-13, Section 4.1.2: Should at least part of this be in the Results section?
*While the thinning plots contain some model results, because the results section focuses on the effect of input parameters on model output as opposed to comparison with data, we would like to leave this section in the discussion. Moving it to the results would require breaking it up amongst a number of different sections and lose the coherence of a single discussion of thinning rates.*

P. 13, lines 423-425: It looks like the SSA simulations thin too little for all three glaciers upstream of their grounding lines.
*This is perhaps a question of what "reasonably well" means, so we have eliminated that phrasing and incorporated this point. The relevant text now reads:* "In general, the shallow-shelf simulations approximately match the pattern of observed surface change downstream of the grounding line, but show too little thinning upstream (Error! Reference source not found.b-d)."

Figure 2: You say "shelf total melt rates are most sensitive to melt rates between ~250 and 600 meters." Why not choose two depths where the PDF values are equal, e.g. ~250 and ~800m?
*Yes, this is probably more appropriate. It has been changed to* "between ~250 and 800m" *where the PDF values are approximately 0.014.*

Table 1: Simulation number 8 has an asterisk indicating that the retreat was entirely forced, but in Figure 5 we can see that the retreat continues beyond the 18y grounding line for Smith glacier.
*It was not our intention for the asterisk to indicate that, but we see how that was unclear from the table caption. We have added an additional note in the table on the simulations that continued retreating beyond the period of explicit forcing. The caption now indicates* "The last column indicates whether the Smith Glacier grounding line retreated over 15 km within the simulation, with starred entries indicating that retreat was explicitly forced. Daggers indicate that some grounding-line retreat continued beyond the period of explicit forcing."

Figure 4: The important first 5 years of grounding line retreat are covered by the plot labels for Smith and Kohler glaciers. Similar for Figure 5.
*Labels have been moved on both figures.*

Technical Corrections

P. 2, lines 49-50: I found this wording confusing – it sounds like "committed" is a verb. Maybe reword it?
*Changed to* "Modeling of the grounded portion of the Smith, Pope, Kohler catchment indicates that these glaciers are committed to further retreat on decadal timescales"

P. 4, line 121: "allows us to"

*Fixed*

P. 5, line 155: "We ran a suite"
*Fixed*

P. 6, line 178: I think this should say something like "...errors due to the assumption are alleviated through choice of the sliding coefficient..."
*This alternative phrasing is a bit overly optimistic. We are confident that there is compensation, but whether that results in a model that is closer to reality is not clear.*

P. 7, line 232: Perhaps a matter of taste, but I like the word "margins" more than "edge." Edge makes me think of the calving front.
*Changed*

P. 8, line 279: 20011 → 2011
*Fixed*

P. 10, line 321: should say "...using all four melt distributions...?"
*Fixed*

P. 10, line 329: I think the referral to Figure 6 is repetitive, given that the whole paragraph is referring to the figure. You could make it a referral to Figures 6c-d if you want to be specific.
*Switched to 6c-d*

P. 11, line 374: mix → mixes
*Fixed*

P. 11, line 379: instantiate → induce (or trigger)?
*Changed to "induce"*

P. 12, line 391: "simulations with marginal"
*Fixed*

P. 12, line 401: indicate → indicates
*Fixed*

P. 12, line 405: 19 → 21
*Good catch, fixed*

P. 14, line 458: across → along?
*Fixed*

P. 14, line 463: "There is a variety..."

*Fixed the missing "a", but left he verb as "are" since both "is" and "are" are grammatical here.*

P. 14, line 470: complicate → complicates
*Fixed*

P. 14, lines 482-483: The way this is worded makes it sound like it was the SSA simulations that required HPC. Maybe say "... allowing the use of local workstations rather than requiring high-performance computing resources."
*Indeed, this phrasing is much clearer.*

P. 15, line 509: have → "has" or "may have"
*Fixed*

P. 15, line 528: "not be an indication" ?
*Changed to "not indicate"*

P. 16, line 531: "we did not have" Figure 1: I assume there should be a box showing the study area on the map of Antarctica – I can't see one on my printed copy.
*Yes, thank you. There seems to have been an error in PDF conversion that we did not catch.*

Figure 3, line 759: "thick for 1Obs, thin for 2Obs."
*Fixed*

Figure 5, line 779: double periods
*Fixed*

Figure 5, line 782-782: "Color of the line indicates year" already stated in line 781.
*Thanks, fixed.*

Figure 6: Axis labels are overlapping for 6a and 6c.
*Fixed.*

Figure 6, line 790: "difference...result" → "difference . . . results" or "differences . . . result
*Corrected to "differences . . . result"*